

# S³GNN: Efficient Global Mixing and Local Message Passing for Long-Range Graph Learning

**Dai Shi** [*1]  **Luke Thompson** [*2]  **Linhan Luo** [*2]  **Lequan Lin** [2]  **Andi Han** [2]
**Junbin Gao** [2]  **José Miguel Hernández Lobato** [1]

## Abstract

Message-passing neural networks (MPNNs) often suffer from an information bottleneck when capturing long-range dependencies, leading to the oversquashing (OSQ) phenomenon. Alongside spatial connectivity enrichment (e.g., rewiring), recent studies have shown that spectral filtering can yield strong long-range learning outcomes, as spectral operators enable global information mixing that alleviates OSQ. These approaches achieve this either by stabilizing the Jacobian energies in deep propagation or by guaranteeing OSQ mitigation under strong theoretical assumptions. We examine the practical attainability of these guarantees and show that the associated Jacobian sensitivity lower bound is generally difficult to achieve in practice. We then propose S³GNN, which mitigates OSQ without such restrictive assumptions by lightweightly reintroducing omitted components with substantially lower computational complexity, while standard stability constraints on feature transformations remain effective under our new dynamics. Extensive experiments across diverse domains (e.g., long-range benchmarks, KGQA, and mesh-based fluid dynamics) demonstrate that S³GNN achieves up to an order-of-magnitude error reduction with up to 50% fewer parameters. Our code can be found in https://github.com/EEthanShi/S3-GNN.git.

## 1. Introduction

Graph neural networks (GNNs) have emerged as the fundamental deep learning models for handling graph-structured

[1]University of Cambridge, Cambridge, UK. [2]University of Sydney, Australia. Correspondence to: Dai Shi <ds2213@cam.ac.uk>.

*Proceedings of the 43rd International Conference on Machine Learning*, Seoul, South Korea. PMLR 306, 2026. Copyright 2026 by the author(s).

data (Wu et al., 2020; Ji et al., 2021; Sato, 2020; Rusch et al., 2023). The oversquashing phenomenon, one of the recently identified problems within GNN dynamics (Shi et al., 2023; Topping et al., 2022; Alon & Yahav, 2021), refers to the degradation of learning outcomes when information from distant nodes cannot be effectively communicated through limited message-passing channels, causing long-range dependencies to be compressed and lost in feature propagations. The mainstream of the research in mitigating the OSQ problem refers to the rewiring techniques, which enrich the graph connectivity based on some graph indicators, such as curvature (Topping et al., 2022; Fesser et al., 2023; Nguyen et al., 2023) and spectral gap (Black et al., 2023; Deac et al., 2022; Karhadkar et al., 2023), to name a few.

Rather than spatially enriching graph connectivity, recent studies (Geisler et al., 2024; Hariri et al., 2025) have illustrated the potential of resolving OSQ with the help of spectral filtering due to its advantage of providing global long-range information. Such spectral methods have achieved excellent performance by (i) stabilizing the weight matrices to prevent vanishing/exploding Jacobian energies in depth, as in Hariri et al. (2025), or by (ii) combining spatial and spectral dynamics, where the theoretical guarantees for OSQ mitigation typically rely on strong conditions (Geisler et al., 2024).

In this paper, we revisit the OSQ vanishing analysis in (Geisler et al., 2024) and verify that the corresponding theoretical lower bound is *difficult to attain* in practice. We then demonstrate that effective alleviation of OSQ can still be achieved even after relaxing these strong conditions by carefully reintroducing previously omitted model components in a lightweight manner, with *significantly lower computational complexity*. Finally, we incorporate stability constraints on the feature transformation into our new model and show that such constraints continue to play a stabilizing role on the Jacobian energy, thereby yielding a simple yet effective architecture for mitigating OSQ in practice. We summarize our contributions as follows.

- **Identification on the gap between theory and practice.** We revisit the OSQ-vanishing analysis of spectral

GNNs, e.g., (Geisler et al., 2024), and show that the associated theoretical lower bound is difficult to attain in practice.

- **A lightweight global-mixing dynamics with non-vanishing influence.** We propose $S^3$GNN, which reintroduces spatial propagation and a simple feature transform on top of a low-rank global mixing term, enabling efficient global information exchange with per-layer complexity comparable to GCN and *without expensive eigendecomposition*; although motivated by spectral filtering, our model is implemented entirely in the spatial domain and admits a distance-independent non-vanishing influence lower bound.

- **Stabilized propagation via constrained feature transforms.** Inspired by stability-constrained spectral GNNs, we incorporate antisymmetric constraints on the feature transformation and show that the resulting Jacobian energy is still stable in our proposed model, yielding a simple yet effective OSQ-mitigating architecture.

- **Strong empirical performance with reduced cost.** We evaluate our proposed model via massive experiments. Our model achieves *orders-of-magnitude* better results on both synthetic and real-world datasets at lower computational cost. Furthermore, we show that our model can serve as a strong backbone across multiple tasks, including knowledge graph question answering (KGQA), topological interpolation, and fluid dynamics prediction.

## 2. Background

In this work, we denote a graph as a tuple $\mathcal{G} = (\mathcal{V}, \mathcal{E})$ where $\mathcal{V}$ and $\mathcal{E}$ denote the set of $N$ nodes and $m$ edges, respectively. In most cases, we assume that $\mathcal{G}$ is undirected, i.e., if $(i,j) \in \mathcal{E}$, then $(j,i) \in \mathcal{E}$. We let $\mathbf{A} \in \mathbb{R}^{N \times N}$ be the unweighted adjacency matrix, and the normalized adjacency matrix is denoted by $\widehat{\mathbf{A}} = \mathbf{D}^{-1/2}\mathbf{A}\mathbf{D}^{-1/2}$ where $\mathbf{D}$ is the degree matrix. We further define $\mathbf{L} \in \mathbb{R}^{N \times N}$ as the graph Laplacian matrix with $\mathbf{L} = \mathbf{D} - \mathbf{A}$. Accordingly, the normalized graph Laplacian is given by $\widehat{\mathbf{L}} = \mathbf{I} - \widehat{\mathbf{A}}$, and has eigendecomposition such that $\widehat{\mathbf{L}} = \mathbf{U}\mathbf{\Lambda}\mathbf{U}^\top$. In this paper, we let $\{(\lambda_i, \mathbf{u}_i)\}_{i=1}^N$ be the set of eigenvalue and eigenvector pairs of $\widehat{\mathbf{L}}$. Finally, a connected component of $\mathcal{G}$ is a maximal subset of nodes $\mathcal{C} \subseteq \mathcal{V}$ such that any two nodes in $\mathcal{C}$ are connected by a path in $\mathcal{G}$. Suppose $\mathcal{G}$ has $k$ connected components $\{\mathcal{C}_r\}_{r=1}^k$ with $|\mathcal{C}_r| = n_r$. It is well known that for unnormalized Laplacian $\mathbf{L}$, the eigenvalue $0$ has multiplicity $k$, and an orthonormal basis of the $0$-eigenspace is given by $\{\mathbf{v}^{(r)}\}_{r=1}^k$, where $(\mathbf{v}^{(r)})_i = \frac{1}{\sqrt{n_r}}$ if $i \in \mathcal{C}_r$, and $0$ otherwise.

### 2.1. Graph Neural Networks

In general, there are two types of GNNs. Spatial message passing neural networks (MPNNs) propagate node features by gathering their *local* neighboring information (Gilmer et al., 2017), resulting in the following dynamic

$$\mathbf{h}_i(\ell+1) = \phi_\ell(\mathbf{h}_i(\ell), \sum_{j \in \mathcal{N}_i} \widehat{\mathbf{A}}_{ij}\, \psi_\ell(\mathbf{h}_i(\ell), \mathbf{h}_j(\ell))), \quad (1)$$

where $\mathbf{h}_i(\ell)$ represents the feature of node $i$ at layer $\ell$, with $\mathbf{h}_i(0) = \mathbf{x}_i$, and $\psi_\ell : \mathbb{R}^{d_\ell} \times \mathbb{R}^{d_\ell} \to \mathbb{R}^{d_\ell}$, $\phi_\ell : \mathbb{R}^{d_\ell} \times \mathbb{R}^{d'_\ell} \to \mathbb{R}^{d_{\ell+1}}$ are channel-mixing and update functions, respectively. One typical example of MPNN is GCN and its variants (Kipf & Welling, 2017), with the dynamic defined as $\mathbf{H}(\ell+1) = \sigma(\widehat{\mathbf{A}}\mathbf{H}(\ell)\mathbf{W}(\ell))$. Another routine of GNN is to conduct feature propagation through *global* spectral filtering in the eigenspace of $\mathbf{L}$. That is

$$\mathbf{H}(\ell+1) = \mathbf{U}(g_\theta(\mathbf{\Lambda}))\mathbf{U}^\top\mathbf{H}(\ell)\mathbf{W}(\ell), \quad (2)$$

where $g_\theta(\cdot) : \mathbb{R} \to \mathbb{R}$ is the learnable spectral filtering function parameterized by $\theta$, and $g_\theta(\mathbf{\Lambda}) = \text{diag}(g_\theta(\lambda_i))$. To avoid the $\mathcal{O}(N^3)$ eigendecomposition in Eq. (2), the Cheb-Net in (Defferrard et al., 2016) uses $\mathbf{U}\, g_\theta(\mathbf{\Lambda})\, \mathbf{U}^\top\mathbf{H} \approx \sum_{p=0}^K \beta_p\, T_p(\widetilde{\mathbf{L}})\, \mathbf{H}$, where $T_p(\cdot)$ denotes the Chebyshev polynomial of degree $p$, $\{\beta_p\}_{p=0}^K$ are the learnable Chebyshev coefficients, and $\widetilde{\mathbf{L}} := \frac{2}{\lambda_{\max}}\mathbf{L} - \mathbf{I}$ is the rescaled Laplacian (with $\lambda_{\max}$ the largest eigenvalue of $\mathbf{L}$).

### 2.2. Oversquashing in GNNs

OSQ was first characterized as an information bottleneck in message passing (Alon & Yahav, 2021) and later quantified via the Jacobian sensitivity score (Topping et al., 2022):

$$\text{OSQ}^{(\ell)}(i, s) = \left\| \frac{\partial \mathbf{h}_i(\ell)}{\partial \mathbf{x}_s} \right\|, \quad (3)$$

which typically decays rapidly with depth $\ell$, limiting long-range communication and degrading performance on tasks that rely on distant interactions (Shi et al., 2023). Motivated by their global filtering behavior, spectral GNNs such as ChebNet (Defferrard et al., 2016), Stable-ChebNet (Hariri et al., 2025), and $S^2$GNN (Geisler et al., 2024) have recently shown strong performance on long-range benchmarks. In particular, Stable-ChebNet follows

$$\mathbf{H}(\ell+1) = \mathbf{H}(\ell) + \epsilon \sum_{p=0}^K T_p(\widetilde{\mathbf{L}})\, \mathbf{H}(\ell)\widehat{\mathbf{W}}_p(\ell), \quad (4)$$

where $\epsilon > 0$ is the step size and $\widehat{\mathbf{W}}_p(\ell)$ is antisymmetric, i.e., $-\widehat{\mathbf{W}}_p(\ell) = \widehat{\mathbf{W}}_p^\top(\ell)$. The dynamic of $S^2$GNN can be written as

$$\mathbf{H}^{(\ell+1)} = \mathbf{U}\Big(g_\theta^{(\ell)}(\mathbf{\Lambda}) \odot \big[\mathbf{U}^\top f_{\widetilde{\theta}}^{(\ell)}(\mathbf{H}^{(\ell)})\big]\Big) + \widehat{\mathbf{A}}\mathbf{H}(\ell)\mathbf{W}(\ell), \quad (5)$$

where $\odot$ denotes the element-wise product and $f_{\widetilde{\theta}}^{(\ell)} : \mathbb{R}^{N \times d_\ell} \to \mathbb{R}^{N \times d_\ell}$ is a feature transformation function. We conduct a literature review on the MPNN methods for mitigating the OSQ problem in the Section 5.

## 3. Method

### 3.1. Gap Between Theory and Practice

In this section, we present the design of our $S^3$GNN model. Based on the description of the OSQ problem aforementioned, to ensure an effective *information transaction* between node pairs (so that the OSQ problem can be diminished), the GNN dynamic shall possess a positive lower bound, i.e., $\left\| \frac{\partial \mathbf{h}_i(\ell)}{\partial \mathbf{x}_s} \right\| > 0$ for any layer $\ell$. Although remarkable conclusions have been explored by delivering upper bounds of $\left\| \frac{\partial \mathbf{h}_i(\ell)}{\partial \mathbf{x}_s} \right\|$ via different dynamics and rewiring methods (Topping et al., 2022; Banerjee et al., 2022; Black et al., 2023), a lower bound is only presented for the dynamic of $S^2$GNN described in Eq. (5). Specifically, consider the dynamic in Eq. (5): if one sets (i) $f_{\widetilde{\theta}}^{(\ell)} = \mathrm{id}$; (ii) $\widehat{\mathbf{A}}\mathbf{H}(\ell)\mathbf{W}(\ell) = 0$; (iii) chooses the spectral filter such that $g_\theta^{(\ell)}(\lambda) = C(\theta) > 0$ when $\lambda = 0$, and $g_\theta^{(\ell)}(\lambda) = 0$ when $\lambda > 0$, where $C(\theta)$ is the positive filtering coefficient determined by the model parameters, then for any $i, s \in \mathcal{C}_r$, we have the lower bound as

$$\left\| \frac{\partial \mathbf{h}_i(\ell)}{\partial \mathbf{x}_s} \right\| \geq \frac{C^\ell(\theta)}{2|\mathcal{E}_{\mathcal{C}_r}|}, \tag{6}$$

where $|\mathcal{E}_{\mathcal{C}_r}|$ is the number of edges in $\mathcal{C}_r$ [1]. One can check that under assumptions (i)-(iii), the $S^2$GNN dynamic becomes the simple diagonal spectral filtering dynamic

$$\begin{aligned} \mathbf{H}(\ell+1) &= \mathbf{U}\,\mathrm{diag}\big(g_\theta^{(\ell)}(\mathbf{\Lambda})\big)\,\mathbf{U}^\top \mathbf{H}(\ell) \\ &= C(\theta)\,\mathbf{U}\,\mathrm{diag}\big(\mathbf{1}_{\{\lambda_i=0\}}\big)\,\mathbf{U}^\top \mathbf{H}(\ell), \end{aligned} \tag{7}$$

which is substantially different from the concrete implementation of $S^2$GNN in Eq. (5). Accordingly, one may want to exploit the following question.

*Can $S^2$GNN really exceed the theoretical lower bound in Eq. (6) if its concrete dynamic is that of Eq. (5)?*

In this spirit, we test the dynamics in Eq. (5) and Eq. (7) in *peptides-struct* dataset, a gold-standard long-range graph benchmark (Dwivedi et al., 2022). Both models are trained via the MAE loss, with the Jacobian norms computed prior to the global pooling layer. We note that given most of the graphs in *peptides-struct* are connected, meaning that

[1]We note that the original lower bound in (Geisler et al., 2024) is defined via entry-wise 1-norm (Theorem 2). Since $\| \cdot \|_2 \geq \frac{1}{d}\| \cdot \|_{L_1}$ for any $d \times d$ matrix, we have Eq. (6).

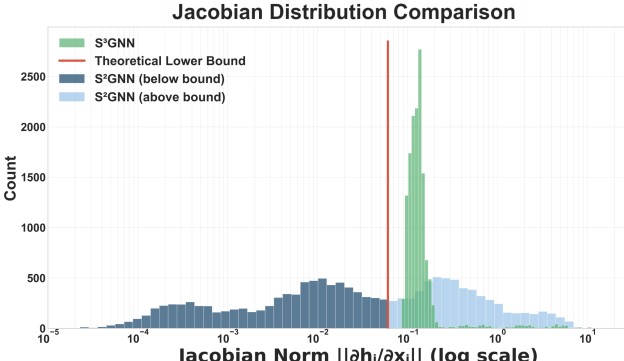

*Figure 1.* Jacobian norm distribution comparison between $S^2$GNN, diagonal filtering in Eq. (7), and our proposed $S^3$GNN.

there is only one 0-eigenvalue to be filtered in Eq. (7). Accordingly, in the non-normalized Laplacian setting, one can verify that $\left\| \frac{\partial \mathbf{h}_i(\ell)}{\partial \mathbf{x}_s} \right\| = \frac{C^\ell(\theta)}{2N}$, i.e., a constant. We followed the exact hyperparameter settings for $S^2$GNN, yielding test MAE over 5 runs as $0.2487 \pm 0.0019$ (33.5s per epoch) and $0.6824 \pm 0.0149$ (0.46s per epoch) for dynamics in Eq. (7). The Jacobian norm distribution (log scale) of one test graph sample (more examples are in Appendix B.1) is shown in Figure 1. We observe many more **below-bound values** than **above-bound values**, suggesting a gap between the idealized theoretical setting and practical implementations , where the *computational complexity* is much higher. For example, for implementing Eq. (5), rather than conducting simple diagonal filtering like in Eq. (7), $g_\theta$ is parameterized by the composition of Gaussian smearing and linear transformation on a specific spectral truncation (approximately 70% of $\mathbf{\Lambda}$ from lowest), and the feature transformation $f_{\widetilde{\theta}}$ is usually implemented with a non-linear gating mechanism.

### 3.2. Lightweight Global Mixing with Non-Vanishing Influence

Having observed the phenomenon in Section 3.1, it is natural for us to explore the following.

*Can one maintain the advantage in Eq. (7) for the OSQ problem, while reintroducing the sacrificed components with significantly lower complexity?*

Accordingly, we consider the following continuous-time dynamics.

$$\frac{\partial \mathbf{H}(t)}{\partial t} = \mathbf{U}\,\mathrm{diag}\big(g_\theta^{(t)}(\mathbf{\Lambda})\big)\mathbf{U}^\top \mathbf{H}(t)\mathbf{W}(t) + \widehat{\mathbf{A}}\mathbf{H}(t)\mathbf{W}(t), \tag{8}$$

where we reintroduce simple feature transformation to a *linear* map, $f_{\widetilde{\theta}}(\mathbf{H}) = \mathbf{H}\mathbf{W}_{\widetilde{\theta}}$, and keep the spatial part in the model dynamics. In addition, we maintain $g_\theta(\lambda) = 0$ for $\lambda > 0$ as in Eq. (7). Applying the forward Euler scheme with step size $\epsilon$ yields the discrete update

$\mathbf{H}(\ell + 1) = \mathbf{H}(\ell) + \epsilon\Big(\mathbf{U}\operatorname{diag}\big(g_\theta^{(\ell)}(\boldsymbol{\Lambda})\big)\mathbf{U}^\top\mathbf{H}(\ell)\mathbf{W}(\ell) + \widehat{\mathbf{A}}\mathbf{H}(\ell)\mathbf{W}(\ell)\Big)$. One can check that this dynamic substantially reduces the computational cost in S$^2$GNN. In addition, one can write $\mathbf{P}_\theta = \mathbf{U}\operatorname{diag}\big(g_\theta(\boldsymbol{\Lambda})\big)\mathbf{U}^\top = \sum_{r=1}^k \alpha_{\theta,r}\, \mathbf{v}^{(r)}\mathbf{v}^{(r)\top}$, where $\{\mathbf{v}^{(r)}\}_{r=1}^k$ are the orthonormal basis vectors of the 0-eigenspace of $\mathbf{L}$, and $(\mathbf{v}^{(r)}\mathbf{v}^{(r)\top})_{ij} = 1/n_r$ for $i, j \in \mathcal{C}_r$ and 0 otherwise. Therefore, the propagation can be implemented **without computing eigendecomposition**. As a result, each layer only involves (i) a *sparse neighborhood aggregation*, and (ii) a *low-rank global mixing*, resulting in

$$\mathbf{H}(\ell + 1) = \mathbf{H}(\ell) + \epsilon\Big(\mathbf{P}_\theta(\ell)\mathbf{H}(\ell)\mathbf{W}(\ell) + \widehat{\mathbf{A}}\mathbf{H}(\ell)\mathbf{W}(\ell)\Big), \tag{9}$$

which shows the computational complexity of the dynamic comparable to classic GCNs. More importantly, for each connected component $\mathcal{C}_r$, one may consider $\mathbf{P}_\theta$ as adding a *scalar-reweighted fully-connected* subgraph, and this setting ensures *effective information transaction* within each connected component. This aligns with the spirit of methods which densify graph connectivity through rewiring (Topping et al., 2022) and graph transformers (Wu et al., 2023a) in mitigating OSQ. Formally, one can derive the following conclusion (proof in Appendix A).

**Proposition 1** (Non-vanishing influence (existence direction))**.** *Consider the dynamics in Eq.* (9)*. Fix a connected component $\mathcal{C}_r$ with $|\mathcal{C}_r| = n_r$. Assume that the feature transform $\mathbf{W}(\ell) \in \mathbb{R}^{d\times d}$ is non-degenerate, i.e., $\sigma_{\min}(\mathbf{W}(\ell)) > 0$ $\forall \ell$, and that the mixing coefficient satisfies $\alpha_{\theta,r}^{(\ell)} > 0$. Then for any $i, s \in \mathcal{C}_r$, layer number $\ell$ such that*

$$\Big\|\frac{\partial \mathbf{h}_i(\ell)}{\partial \mathbf{x}_s}\Big\| \geq \prod_{p=0}^{\ell-1}\Big(\varepsilon\,\frac{\alpha_{\theta,r}^{(p)}}{n_r}\,\sigma_{\min}\big(\mathbf{W}(p)\big)\Big). \tag{10}$$

*In particular, the lower bound is independent of the graph distance between $i$ and $s$ within $\mathcal{C}_r$.*

Proposition 1 further indicates that to engage long-range communication between nodes when it is highly needed, one may expect to have $\alpha_{\theta,r} > 0$ and larger when $\ell$ goes higher. We frequently observe this phenomenon in our numerical experiments, e.g., see more results in Figure. 3.

In addition, the functionality of $\alpha$ shares a higher-level motivation compared to the recently developed virtual nodes (VN) methods in GNN (Gilmer et al., 2017; Rosenbluth et al., 2024; Southern et al., 2025; Qian et al., 2024; Choi et al., 2026) where node feature information is propagated (e.g., averaged) through artificial argument nodes. We include empirical comparisons with VN baselines in Section 4.2.

### 3.3. Jacobian Stability and Flexible Choice of Feature Transformation

With the increase in layers, the dynamics can become increasingly unstable without properly constraining the weights. Specifically, the dynamics in Eq. (9) can suffer from the unstable Jacobian energy in each layer (Hariri et al., 2025). To see this, one can let $\mathbf{J}(\ell) = \frac{\partial \operatorname{vec}(\mathbf{H}(\ell+1))}{\partial \operatorname{vec}(\mathbf{H}(\ell))} \in \mathbb{R}^{Nd\times Nd}$ (assume $\mathcal{G}$ is connected for simplicity), so that the Jacobian energy of each $\ell$ can be denoted as $\|\mathbf{J}(\ell)\|_2$. Motivated by the Stable-ChebNet (Hariri et al., 2025), we incorporate antisymmetric constraint on $\mathbf{W}$, resulting in

$$\mathbf{H}(\ell + 1) = \mathbf{H}(\ell) + \epsilon\Big(\mathbf{P}_\theta(\ell)\mathbf{H}(\ell)\widehat{\mathbf{W}}(\ell) + \widehat{\mathbf{A}}\mathbf{H}(\ell)\widehat{\mathbf{W}}(\ell)\Big), \tag{11}$$

where $\widehat{\mathbf{W}}$ satisfies the antisymmetric property as Eq. (4). In practice, one can deploy two antisymmetric matrices, with each corresponding to the global mixing and spatial parts. As our model was originally motivated by S$^2$GNN (Geisler et al., 2024), but with a simpler structure and a stability guarantee (See proposition 2 below), we therefore name our model as **S$^3$GNN**, and we show the stability guarantee as follows.

**Proposition 2** (Stability)**.** *The Jacobian energy of S$^3$GNN in Eq.* (11) *is stable, i.e., no exponential growth or decay in each layer. That is $\|\mathbf{J}(\ell)\|_2 = 1 + \mathcal{O}(\epsilon^2)$.*

Figure 2 visualizes the propagation of our S$^3$GNN on the Barbell graph, a synthetic node-level regression task for testing OSQ robustness (Bamberger et al., 2025; Hariri et al., 2025). In addition, we show comparisons in MSE loss and number of parameters between our model and Stable-ChebNet. Our key observations are

1. S$^3$GNN delivers **orders-of-magnitude** better MSE with much **fewer parameters**. For example, our model is around 30 times better in the case when $N = 50$ compared to Stable-ChebNet, i.e., 0.17 vs. 0.005, whilst halving the number of parameters, i.e., 262K vs. 591K.

2. Our model achieves this remarkable result with only 4 layers compared to ChebNet and Stable-ChebNet, in which the polynomial order is up to 10 with 4 layers, suggesting a **more effective information transaction**.

3. When parameter count is matched (i.e., 657K), S$^3$GNN maintains leading performances across domains, supporting the effectiveness of incorporating antisymmetric feature transformation in our model.

Lastly, we also present the Jacobian norm distribution of S$^3$GNN results (MAE = 0.2429±0.0014) in Figure 1, and one can observe that all values in **green** are beyond the theoretical lower bound. This directly verifies that the elements

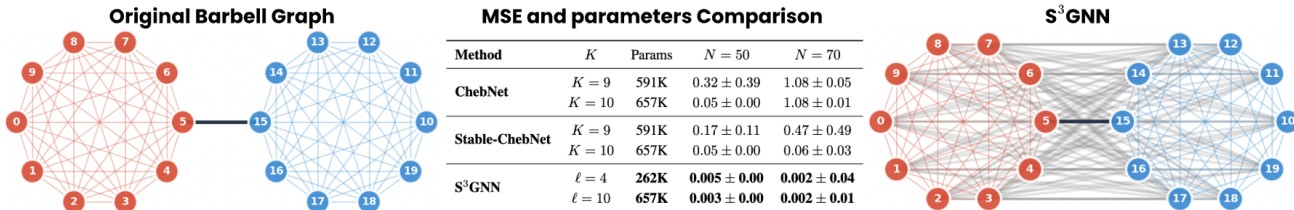

*Figure 2.* Illustration on S$^3$GNN dynamic, i.e., neighborhood aggregation and global mixing in Barbell graph, and comparison between S$^3$GNN with spectral methods, e.g., ChebNet and Stable-ChebNet in terms of learning accuracy and number of parameters. Our S$^3$GNN shows order-of-magnitude better performances with less parameters.

that S$^3$GNN leveraged additional to the diagonal spectral filtering dynamic in Eq. (5) indeed improve the long-range communication between distant nodes from both theoretical (i.e., Proposition 1) and practical sides.

**Flexible Choice of Feature Transform**  In our proposed S$^3$GNN, we constrain $\widehat{\mathbf{W}}$ to be antisymmetric to stabilize the Jacobian energy, $\|\mathbf{J}(\ell)\|_2$. However, the choice of feature transformation can be varied. For example, based on Proposition 1, one may let $\mathbf{W}$ be orthogonal matrix, so that $\sigma_{\min} = 1$ to eliminate the effect of $\mathbf{W}$ in Eq. (10). From a geometric viewpoint, such an orthogonality requirement can be understood as a manifold constraint that restricts $\mathbf{W}$ to lie on the orthogonal group, thereby enforcing norm-preserving feature transport across layers. In addition, inspired by manifold-constrained mixing mechanisms in deep residual architectures (Xie et al., 2025; Mishra, 2026; Shi et al., 2021), one may also consider using doubly stochastic transforms to mix feature channels while preserving simple aggregate quantities (e.g., keeping the per-layer feature mean unchanged). We explore this further in Section 4.6.

## 4. Experiments

We evaluate our S$^3$GNN in multiple tasks, such as graph-level property prediction, large-scale node classification (both in Section 4.1), long-range graph benchmarks (Section 4.2), multi-hop/entity KGQA (Section 4.3), topological interpolation (Section 4.4), and mesh-based fluid dynamics prediction (Section 4.5). In addition, we also conduct ablation studies and provide a discussion on future research directions in Section 4.6. Additional experiments (e.g., on heterophily graphs) and comparisons are included in the Appendix. All experiments are executed on an NVIDIA® H200 SXM GPU (141 GB HBM3e) within an HPC cluster.

### 4.1. Graph Property Prediction Dataset

We first evaluate the performance of S$^3$GNN via the graph property prediction dataset provided by (Corso et al., 2020) followed by the experiment settings in (Gravina et al., 2023; Hariri et al., 2025). The dataset consists of undirected graphs sampled from a diverse collection of random and structured graph families (e.g., Erdős–Rényi,

*Table 1.* Mean test set $\log_{10}(\text{MSE})$ and standard deviation on the Graph Property Prediction dataset. The lower the better.

| Model | Diameter | SSSP | Eccentricity |
|---|---|---|---|
| GCN | $0.7424 \pm 0.0466$ | $0.9499 \pm 0.0001$ | $0.8468 \pm 0.0028$ |
| GAT | $0.8221 \pm 0.0752$ | $0.6951 \pm 0.1499$ | $0.7909 \pm 0.0222$ |
| GraphSAGE | $0.8645 \pm 0.0401$ | $0.2863 \pm 0.1843$ | $0.7863 \pm 0.0207$ |
| GIN | $0.6131 \pm 0.0990$ | $-0.5408 \pm 0.4193$ | $0.9504 \pm 0.0007$ |
| DGC | $0.6028 \pm 0.0050$ | $-0.1483 \pm 0.0231$ | $0.8261 \pm 0.0032$ |
| GRAND | $0.6715 \pm 0.0490$ | $-0.0942 \pm 0.3897$ | $0.6602 \pm 0.1393$ |
| A-DGN | $0.2271 \pm 0.0804$ | $-1.8288 \pm 0.0607$ | $0.7177 \pm 0.0345$ |
| ChebNet | $-0.1517 \pm 0.0343$ | $-1.8519 \pm 0.0539$ | $-1.2151 \pm 0.0852$ |
| Stable-ChebNet | $-0.2477 \pm 0.0526$ | $-2.2111 \pm 0.0160$ | $-2.1043 \pm 0.0766$ |
| S$^2$GNN | $-0.1824 \pm 0.0329$ | $-2.103 \pm 0.0386$ | $-1.4622 \pm 0.0815$ |
| DIFFormer | $-0.2046 \pm 0.0358$ | $-1.9033 \pm 0.0839$ | $-1.4010 \pm 0.0789$ |
| **S$^3$GNN (ours)** | $\mathbf{-0.3301 \pm 0.0342}$ | $\mathbf{-2.6311 \pm 0.0171}$ | $\mathbf{-2.4801 \pm 0.0239}$ |

Barabási–Albert, and caterpillar graphs), thereby covering a wide range of topological properties. Followed by (Gravina et al., 2023), the node number in each graph is between 25 and 35 to increase the task complexity and raising the need for long-range information propagation. Specifically, *Diameter* is a graph-level regression task, whereas *Single-Source Shortest Path (SSSP)* and *Eccentricity* are node-level regression tasks, respectively. In addition, as S$^3$GNN provides a scalar-reweighted fully-connected subgraph for each connected component, we also compare S$^3$GNN with graph transformer-based models, e.g., DIFFormer (Wu et al., 2023a), in which the feature dynamic is conducted via fully-connected graphs. Finally, we also compare our model with S$^2$GNN as shown in Eq. (5).

**Results and Computational Complexity**  The results are contained in Table 1. One can check that S$^3$GNN consistently outperforms all baselines. Especially on the most challenging task, e.g., *Eccentricity*, S$^3$GNN yields an *order-of-magnitude* improvement compared to S$^2$GNN, while consistently improving performance on *Diameter* and *SSSP*. In addition, unlike spectral methods like ChebNet and Stable-ChebNet, which require 4 to 6 layers and a 4th-order polynomial to invoke the best results, our model achieves the best results with only 4 layers and a lower computational cost. For example, assume $\mathcal{G}$ is connected, the global mixing term in S$^3$GNN reduces to a mean aggregation and can be computed in $\mathcal{O}(Nd)$, leading to a per-layer complexity $\mathcal{O}(|\mathcal{E}|d + Nd^2)$, whereas the complexity for both ChebNet and Stable-ChebNet is $\mathcal{O}(K|\mathcal{E}|d + Nd^2)$, roughly a factor-$K$ higher propagation cost per layer. As aforementioned,

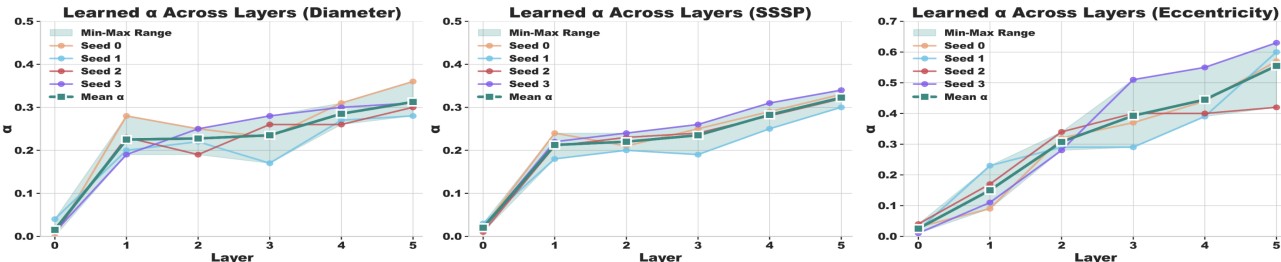

*Figure 3.* Changes of learned $\alpha$ values across the layers.

*Table 2.* Node classification accuracy on OGB datasets.

*(a)* Accuracy on ogbn-arxiv.

| Model | ogbn-arxiv |
|---|---|
| GCN | $71.74 \pm 0.29$ |
| ChebNet | $73.27 \pm 0.23$ |
| ChebNetII | $72.32 \pm 0.23$ |
| GraphSAGE | $71.49 \pm 0.27$ |
| GAT | $72.01 \pm 0.20$ |
| NodeFormer | $59.90 \pm 0.42$ |
| GraphGPS | $70.92 \pm 0.04$ |
| GOAT | $72.41 \pm 0.40$ |
| EXPHORMER+GCN | $72.44 \pm 0.28$ |
| SPEXPHORMER | $70.82 \pm 0.24$ |
| DIFFormer | $72.21 \pm 0.33$ |
| Stable-ChebNet | $75.42 \pm 0.28$ |
| $S^2$GCN | $73.89 \pm 0.58$ |
| $S^3$GNN (ours) | $\mathbf{76.38 \pm 0.32}$ |

*(b)* Accuracy on ogbn-proteins.

| Model | ogbn-proteins |
|---|---|
| MLP | $72.04 \pm 0.48$ |
| GCN | $72.51 \pm 0.35$ |
| ChebNet | $77.55 \pm 0.43$ |
| SGC | $70.31 \pm 0.23$ |
| GCN-NSAMPLER | $73.51 \pm 1.31$ |
| GAT-NSAMPLER | $74.63 \pm 1.24$ |
| SIGN | $71.24 \pm 0.46$ |
| NodeFormer | $77.45 \pm 1.15$ |
| SGFormer | $79.53 \pm 0.38$ |
| SPEXPHORMER | $80.65 \pm 0.07$ |
| DIFFormer | $77.69 \pm 0.25$ |
| Stable-ChebNet | $79.89 \pm 0.18$ |
| $S^2$GCN | $80.58 \pm 0.72$ |
| $S^3$GNN (ours) | $\mathbf{81.90 \pm 0.12}$ |

the propagation of $S^2$GNN requires eigendecomposition, which incurs an additional computational cost (e.g., $\mathcal{O}(N^3)$ in the worst case) and makes it less scalable to large graphs.

$\alpha$ **Dynamics** Recall that one assumption in our theory is the mixing coefficient $\alpha_\theta^{(\ell)} > 0, \ \forall \ell$, and one expects $\alpha_\theta^{(\ell)}$ to become larger when $\ell$ is larger, where long-range communication between nodes is highly required. Interestingly, we empirically observed that the positive assumption on $\alpha$ can be **automatically satisfied**, evidenced by Figure 1 where one learnable $\alpha$ leads to a positive theoretical lower bound and Figure 3 in which an increasing trend between $\alpha_\theta$ and $\ell$ is consistently observed for all datasets. These findings indicate that the positivity and growth of $\alpha_\theta^{(\ell)}$ emerge naturally during training, rather than being enforced explicitly, thereby aligning the learned dynamics with the conditions required by our theoretical analysis. Lastly, $\alpha$ does not require delicate tuning in practice: it is learned over with a maximum of 3 values of initialization, and $S^3$GNN follows the same search space as Stable-ChebNet for shared hyperparameters without introducing additional task-specific tuning procedures, see Section B.3 for more details.

**Scalability over Large-Scale Datasets** In addition to the graph-level tasks in graph property prediction datasets, we further evaluate the performance of $S^3$GNN in node classification tasks via real-world large-scale datasets, e.g., *ogbn-arxiv and ogbn-proteins*. The results can be found in Ta-

*Table 3.* Long-range benchmark results.

| Model Type | Model | peptides-func (AP↑) | peptides-struct (MAE↓) |
|---|---|---|---|
| *Transformer* | SAN+LapPE | $63.84 \pm 1.21$ | $0.2683 \pm 0.0043$ |
| | TIGT | $66.79 \pm 0.74$ | $0.2485 \pm 0.0015$ |
| | Specformer | $66.86 \pm 0.64$ | $0.2550 \pm 0.0014$ |
| | Exphormer | $65.27 \pm 0.43$ | $0.2481 \pm 0.0007$ |
| | G.MLPMixer | $69.21 \pm 0.54$ | $0.2475 \pm 0.0015$ |
| | Graph ViT | $69.42 \pm 0.75$ | $0.2449 \pm 0.0016$ |
| | GRIT | $69.88 \pm 0.82$ | $0.2460 \pm 0.0012$ |
| *Rewiring* | LASER | $64.40 \pm 0.10$ | $0.3043 \pm 0.0019$ |
| | DRew-GCN | $69.96 \pm 0.76$ | $0.2781 \pm 0.0028$ |
| | +PE | $71.50 \pm 0.44$ | $0.2536 \pm 0.0015$ |
| *State Space* | Graph Mamba | $67.39 \pm 0.87$ | $0.2478 \pm 0.0016$ |
| | GMN | $70.71 \pm 0.83$ | $0.2473 \pm 0.0025$ |
| | MP-SSM | $69.93 \pm 0.52$ | $0.2458 \pm 0.0017$ |
| *Virtual/Fractal Nodes* | GCN+VN | $65.40 \pm 0.66$ | $0.2509 \pm 0.0030$ |
| | GatedGCN+VN | $68.10 \pm 0.33$ | $0.2498 \pm 0.0042$ |
| | GCN+FN | $67.80 \pm 0.54$ | $0.2456 \pm 0.0019$ |
| | GatedGCN+FN | $69.50 \pm 0.47$ | $0.2470 \pm 0.0039$ |
| *GNN* | A-DGN | $59.75 \pm 0.44$ | $0.2874 \pm 0.0021$ |
| | ChebNet | $69.61 \pm 0.33$ | $0.2627 \pm 0.0033$ |
| | GCN | $68.60 \pm 0.50$ | $0.2460 \pm 0.0007$ |
| | GRAND | $57.89 \pm 0.62$ | $0.3418 \pm 0.0015$ |
| | GraphCON | $60.22 \pm 0.68$ | $0.2778 \pm 0.0018$ |
| | SWAN | $67.51 \pm 0.39$ | $0.2485 \pm 0.0009$ |
| | $S^2$GNN | $72.75 \pm 0.66$ | $0.2487 \pm 0.0019$ |
| | +PE | $73.11 \pm 0.66$ | $0.2447 \pm 0.0032$ |
| | Stable-ChebNet | $70.32 \pm 0.26$ | $0.2542 \pm 0.0030$ |
| | $S^3$GNN | $\mathbf{73.20 \pm 0.14}$ | $\mathbf{0.2429 \pm 0.0014}$ |

ble 2, where $S^3$GNN shows outstanding results compared to multiple baselines. In particular, compared to those graph transformer-based methods, such as SGFormer (Wu et al., 2023b), Spexphormer (Shirzad et al., 2024), and DIFFormer (Wu et al., 2023a). This indicates that, to account for long-range interactions between distant nodes, one may simply need to add a scalar-adjusted, low-rank, global-mixing process as in $S^3$GNN, rather than employing a complex attention mechanism between any pair of nodes.

### 4.2. Long-Range Graph Benchmarks

Long-range graph benchmarks (LRGB) (Dwivedi et al., 2022) are designed to specifically test GNN performance on tasks where distant node information is important. We select *peptides-func* and *peptides-struct*, with the former as a graph classification evaluated by average precision (AP), and the latter as graph regression evaluated by mean absolute error (MAE). The results are presented in Table 3, where $S^3$GNN shows leading performance compared to several baseline architectures. Interestingly, we did not observe strong performance from spatial rewiring methods such as LASER (Barbero et al., 2024) and DRew (Gutteridge et al.,

*Table 4.* Synthetic long-range diagnostic results. We report mean performance and standard deviation over five runs.

| Dataset | Model | Performance |
|---|---|---|
| LR-CLUSTER | $S^2$GNN ($\ell = 3$) | $42.1 \pm 1.02$ |
| | $S^2$GNN ($\ell = 6$) | $61.3 \pm 0.47$ |
| | $S^2$GNN ($\ell = 12$) | $81.6 \pm 0.78$ |
| | $S^3$GNN ($\ell = 3$) | $\mathbf{55.3 \pm 0.90}$ |
| | $S^3$GNN ($\ell = 6$) | $\mathbf{64.8 \pm 0.86}$ |
| | $S^3$GNN ($\ell = 12$) | $\mathbf{84.2 \pm 0.65}$ |
| Oversquashing-Extend | $S^2$GNN | $95.1 \pm 0.75$ |
| | $S^3$GNN | $\mathbf{96.9 \pm 0.33}$ |

2023). Instead, GNNs that propagate feature information globally, such as $S^2$GNN and Stable-ChebNet, consistently outperform other baselines. However, compared to Stable-ChebNet, the learning accuracy gain of $S^2$GNN may be due to the additional spatial part and the positional encoding (PE), which requires high-cost eigendecomposition of every input graph. On the other hand, our $S^3$GNN outperforms Stable-ChebNet without conducting eigendecomposition and PE. This suggests that explicitly enhancing global information exchange is a more effective strategy for these long-range graph tasks, and $S^3$GNN achieves this advantage with a *lightweight design and low computational overhead*.

**Synthetic long-range diagnostics**   To further evaluate long-range reasoning beyond the LRGB, we also follow the synthetic diagnostic settings used in $S^2$GNN (Geisler et al., 2024). We consider LR-CLUSTER, which contains 12,000 graphs with an average of 896 nodes, and an oversquashing-extend benchmark consisting of 730 Clique-Path and Ring graphs with an average of 43.8 nodes. As shown in Table 4, $S^3$GNN outperforms $S^2$GNN across all LR-CLUSTER depth settings and also improves performance on the oversquashing-extend benchmark. These results provide additional evidence that $S^3$GNN can capture long-range interactions and mitigate OSQ beyond the peptide benchmarks.

### 4.3. Multi-Hop/Entity KGQA

Knowledge Graph Question Answering (KGQA) aims to answer natural language questions by reasoning over structured knowledge graphs composed of entities and relations (Pan et al., 2024). While recent LLMs have shown strong performance on simple factual queries, they often struggle when questions require reasoning over multiple hops or involve multiple query entities, due to the combinatorial expansion of the graph context and the difficulty of reliably identifying relevant relational paths. As a result, multi-hop and multi-entity questions are regarded as among the most challenging settings in KGQA (Mavromatis & Karypis, 2022).

Following recent work on deploying GNNs for effective

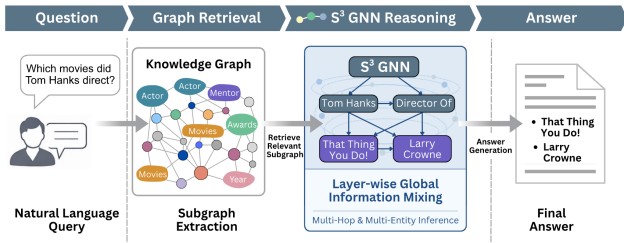

*Figure 4.* Illustration of multi-hop/entity KGQA with $S^3$GNN.

*Table 5.* Performance on multi-hop and multi-entity questions.

| Method | WebQSP (F1) | | CWQ (F1) | | MetaQA-3 (H@1) |
|---|---|---|---|---|---|
| | multi-hop | multi-entity | multi-hop | multi-entity | multi-hop |
| LLM (No RAG) | 48.4 | 61.5 | 33.7 | 32.3 | 29.7 |
| Reasoning GNN | 58.8 | 70.4 | 57.7 | 54.2 | 98.6 |
| Reasoning $S^3$GNN | **69.6** | **73.0** | **68.8** | **61.1** | **98.8** |
| **Improvement** | +10.8% | +2.6% | +11.1% | +6.9% | +0.2% |
| RoG | 63.3 | 65.1 | 59.3 | 58.3 | 84.8 |
| SubgraphRAG | 65.8 | 54.9 | 55.8 | 52.3 | – |
| Reasoning GNN-RAG | 69.8 | 82.3 | 68.2 | 64.8 | 98.6 |
| Reasoning $S^3$GNN-RAG | **70.3** | **85.5** | **68.8** | **64.9** | **98.9** |
| **Improvement** | +0.5% | +3.2% | +0.6% | +0.1% | +0.3% |

graph retrieval (Mavromatis & Karypis, 2024), we study KGQA under a retrieval-augmented generation (RAG) setting, where the model (e.g., Reasoning GNN (Mavromatis & Karypis, 2022)) first retrieves a compact subgraph relevant to the question and then performs reasoning on the retrieved evidence. In this experiment, GNN-based retrieval is particularly focused on multi-hop and multi-entity queries (hops/entities $\geq$ 2), and we replace the original message-passing dynamics of the reasoning GNN with $S^3$GNN, while leaving the remaining RAG pipeline unchanged. We evaluate on the multi-hop and multi-entity subsets of *WebQSP* and *CWQ*, and the multi-hop split of *MetaQA-3*, and report the corresponding F1/H@1 metrics. Figure 4 illustrates the multi-hop/entity KGQA process and the role of $S^3$GNN.

**Results**   From the results in Table 5, $S^3$GNN consistently improves performance in both the presence and absence of RAG. We further observe that, in some cases, the gains obtained by adopting $S^3$GNN are sufficient to surpass the performance of a reasoning GNN equipped with RAG (e.g., 68.8 V.S 68.2 in *CWQ* multi-hop F1 score). This suggests that a more suitable GNN dynamics in the reasoning module may partially substitute explicit retrieval mechanisms and lead to stronger QA performance. The relatively smaller improvement in the RAG setting is also expected, since we only replace the reasoning GNN while keeping the rest of the pipeline unchanged, so gains in the reasoning module alone may not translate proportionally to full-pipeline improvements. Moreover, retrieval partly mitigates the OSQ bottleneck by pre-selecting a compact question-relevant subgraph, reducing the additional gain that $S^3$GNN's global mixing can provide. We leave a deeper study of this interaction for future work.

*Table 6.* Overall 1 (left) and square root (right) 2-Wasserstein distances on brain signals, lower is better.

| Method | Brain signals | |
|---|---|---|
| TSB-BM-GCN | $7.51 \pm 0.08$ | $5.51 \pm 0.06$ |
| TSB-VE-GCN | $7.59 \pm 0.05$ | $5.55 \pm 0.04$ |
| TSB-VP-GCN | $7.67 \pm 0.11$ | $5.64 \pm 0.09$ |
| TSB-BM-S$^3$GNN | $\mathbf{7.25 \pm 0.05}$ | $\mathbf{5.29 \pm 0.18}$ |
| TSB-VE-S$^3$GNN | $\mathbf{7.22 \pm 0.03}$ | $\mathbf{5.24 \pm 0.10}$ |
| TSB-VP-S$^3$GNN | $\mathbf{7.29 \pm 0.05}$ | $\mathbf{5.30 \pm 0.05}$ |

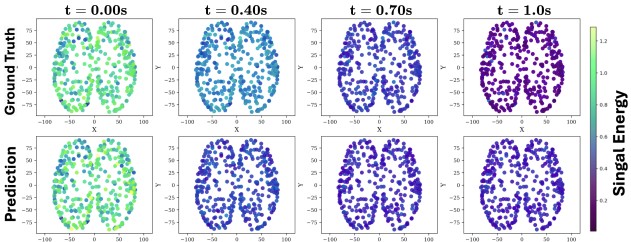

*Figure 5.* **Interpolation results of TSB-VE-S$^3$GNN.** Our model successfully produces the intermediate energy distributions between the initial and ending brain signals.

## 4.4. Topological Interpolation

In this experiment, we show that S$^3$GNN can serve as a strong backbone for topological interpolation. Specifically, we consider the topological Schrodinger bridge matching (TSBM) model proposed in (Yang, 2025), in which we aim to predict the intermediate-energy distributions between high- and low-energy brain signals (Van Essen et al., 2013). The training of TSBM is conducted by using an alternating scheme and both forward and backward processes governed by the graph stochastic heat diffusion equation $dX_t = -cLX_t dt + g_t dW_t$. Fixing $X_T \sim p_T$ (prior) and $X_0 \sim p_0$ (posterior), TSBM seeks to fit two GNNs $\phi_\sharp$ and $\phi_\square$ for forward and backward SDEs for brain signals,

$$dX_t = (-cLX_t + g_t^2 \nabla_x \log \phi_\sharp(t)(X_t))dt + g_t dW_t.$$
(12)

Here, $\nabla_x$ is the gradient of the feature vectors and the backward SDE can be obtained by replacing $\phi_\sharp$ with $\phi_\square$. Our motivation for deploying S$^3$GNN to this task is grounded in neuroscience evidence that transitions of brain signals across different energy states are governed by large-scale functional connectivity and long-range interactions among distant regions, thereby necessitating explicit global information exchange during propagation (Greicius et al., 2003; Bassett & Bullmore, 2006). Accordingly, one would expect GNNs that propagate node features in a global manner to be a better model in terms of approximating the score functions (e.g., $\nabla_x \log \phi_\sharp(t)(X_t)$ ) in Eq. (12). The results are presented in Table 6 and Figure 5. Rather than observing that S$^3$GNN assisted TSB models consistently outperform their

*Table 7.* Results (RMSE) on Cylinder Flow, the lower is better.

| Metric | MGN | S$^3$MGN |
|---|---|---|
| Min Test Loss | 0.173 | **0.102** (41%↓) |
| Min Train Loss | 0.138 | **0.067** (51%↓) |
| Min Velocity RMSE | 0.000589 | **0.000333** (43%↓) |

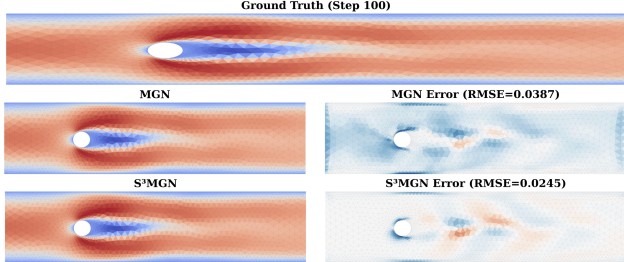

*Figure 6.* **Cylinder flow visualization (Step 100).** Top: ground-truth velocity field. Middle/Bottom: predictions from MGN and S$^3$MGN with corresponding error maps.

GCN counterparts, one can also find that the differences between three types of TSB models, i.e., standard Brownian motion (BM), variance exploding (VE) and variance preserving (VP), are relatively smaller than the performance gap induced by replacing the GCN backbone with S$^3$GNN. We conjecture that the improvements are related to the explicit global information exchange in S$^3$GNN, which is consistent with the long-range interactions in brain signals. This points to a promising direction of integrating task-suitable GNN backbones with (topological) generative modeling frameworks for different data domains.

## 4.5. Mesh-based Fluid Dynamics Prediction

We evaluate S$^3$GNN as a backbone for mesh-based fluid prediction on the cylinder-flow benchmark (Pfaff et al., 2020). At each discrete time step, the spatially discretized flow field is represented as a mesh graph with fixed topology, where node features encode the instantaneous physical states. Given the state at time $t$, the model predicts per-node acceleration (or velocity increment) $\hat{a}_t$, and we obtain the next-step velocity by explicit integration: $\hat{v}_{t+1} = v_t + \Delta t \cdot \hat{a}_t$. Our S$^3$MeshGraphNet (S$^3$MGN) follows the standard MeshGraphNet (MGN) encoder–processor–decoder pipeline, but replaces the processor blocks with S$^3$GNN layers. This design is motivated by the long-range couplings in cylinder wakes, where effective coordination between distant mesh regions is beneficial for stable rollout prediction.

We evaluate both one-step prediction and multi-step rollouts by repeatedly applying the learned dynamics. The quantitative comparison is reported in Table 7, where S$^3$MGN consistently improves over the MGN baseline across training/test objectives and velocity RMSE. In addition, Figure 6 visualizes a long-horizon rollout (Step 100), where S$^3$MGN yields a lower RMSE and noticeably reduced errors in the

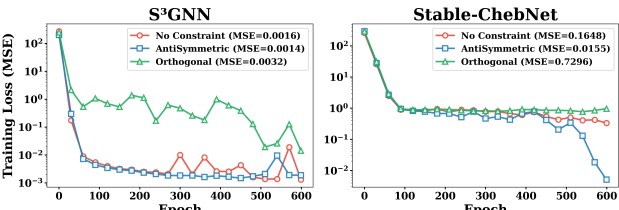

*Figure 7.* Ablation study in Barbell graph with different constraints.

wake region compared to MGN.

### 4.6. Ablation and Future Studies

Finally, we conduct ablation studies on constraints imposed on $\mathbf{W}$, including a free $\mathbf{W}$, an antisymmetric $\widehat{\mathbf{W}}$ (as in $S^3$GNN), and an orthogonal $\mathbf{W}$, where the latter is expected to further tighten the bound of $\left\|\partial \mathbf{h}_i(\ell)/\partial \mathbf{x}_s\right\|$ as discussed in Section 3.3. As shown in Figure 7, both Stable-ChebNet and $S^3$GNN achieve the best performance with the antisymmetric constraint. Notably, the gap between free $\mathbf{W}$ and antisymmetric $\widehat{\mathbf{W}}$ is much smaller in $S^3$GNN than in Stable-ChebNet, suggesting that the benefit of constraining $\mathbf{W}$ depends on the propagation mechanism (e.g., spatial vs. spectral) and the task. Moreover, while orthogonality can more strongly tighten the bound of $\left\|\partial \mathbf{h}_i(\ell)/\partial \mathbf{x}_s\right\|$, it also substantially shrinks the feasible space of $\mathbf{W}$, which may hinder task-specific adaptation and lead to a more unstable optimization behavior. In contrast, antisymmetry appears to regulate Jacobian variation in a more parameter-efficient way, yielding a better stability–expressivity trade-off in this setting. These observations motivate future work on selecting the appropriate degree of constraint for different propagation schemes and long-range learning tasks.

### 5. Related Works and Discussion

**OSQ Mitigation Methods**   Oversquashing arises when information from exponentially many distant nodes is compressed into fixed-dimensional node representations, and has been recognized as a fundamental limitation of message-passing GNNs (Topping et al., 2022; Shi et al., 2023; Alon & Yahav, 2021). Existing mitigation methods can be broadly categorized into three lines. (1) *Spatial rewiring* directly modifies the graph connectivity (i.e., the adjacency) by adding shortcut edges, typically connecting nodes within an $r$-hop neighborhood (Topping et al., 2022; Giraldo et al., 2022). (2) *Spectral rewiring* aims to add edges so as to improve global expansion-related properties of the graph, such as the spectral gap (Karhadkar et al., 2023), total effective resistance (Black et al., 2023), and commute time (Di Giovanni et al., 2024). (3) *Implicit rewiring* constructs better effective connectivity (e.g., denser, reweighted, or fully connected graphs) as a by-product of architectures originally motivated by other considerations, including graph transformers and their variants (Ying et al., 2021; Wu et al.,

2022; 2023a), diffusion-based MPNNs (Thorpe et al., 2022; Chamberlain et al., 2021a; Toth et al., 2022), and multi-track feature propagation methods (Pei et al., 2024; Choi et al., 2024a). We refer the recent survey and positional paper on OSQ problem for more details (Shi et al., 2023; Arnaiz-Rodriguez & Errica, 2026).

### 6. Conclusion

In this work, we identified a gap between recent theoretical OSQ-mitigation guarantees and their practical attainability in graph neural networks. To address this issue, we proposed $S^3$GNN, a lightweight global–local propagation framework that achieves effective long-range information mixing without relying on restrictive spectral assumptions or expensive eigendecomposition. We further showed that standard stability constraints on feature transformations remain effective under the proposed dynamics, continuing to stabilize the Jacobian energy across depth. Extensive experiments across diverse domains demonstrate that $S^3$GNN delivers strong long-range performance with substantially reduced computational and parameter costs.

### Impact Statement

This paper presents work whose goal is to advance the field of machine learning, specifically improving the efficiency and effectiveness of graph neural networks for long-range learning tasks. There are many potential societal consequences of our work, none of which we feel must be specifically highlighted here.

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

# A. Theoretical Proofs

*Proof of Proposition 1.* Fix a connected component $\mathcal{C}_r$ with $|\mathcal{C}_r| = n_r$ and take any $i, s \in \mathcal{C}_r$. For convenience, write the layer operator in Eq. (9) as

$$\mathbf{H}(\ell + 1) = \mathbf{M}(\ell)\,\mathbf{H}(\ell)\,\mathbf{W}(\ell), \qquad \mathbf{M}(\ell) := \widehat{\mathbf{A}} + \mathbf{P}_\theta(\ell).$$

By construction of the 0-eigenspace basis $\{\mathbf{v}^{(r)}\}$ on connected components, $\mathbf{P}_\theta(\ell)$ is block-constant on $\mathcal{C}_r$, hence for all $u, v \in \mathcal{C}_r$,

$$\big(\mathbf{P}_\theta(\ell)\big)_{uv} = \frac{\alpha_{\theta,r}^{(\ell)}}{n_r}, \qquad \big(\mathbf{P}_\theta(\ell)\big)_{uv} = 0 \text{ if } u \notin \mathcal{C}_r \text{ or } v \notin \mathcal{C}_r. \tag{13}$$

In particular, $\big(\mathbf{M}(\ell)\big)_{uv} \geq \big(\mathbf{P}_\theta(\ell)\big)_{uv} \geq 0$.

Let $\mathbf{h}_i(\ell) \in \mathbb{R}^d$ denote the feature of node $i$ at layer $\ell$ (a row vector), and define $\mathbf{J}_{is}(\ell) := \frac{\partial \mathbf{h}_i(\ell)}{\partial \mathbf{x}_s} \in \mathbb{R}^{d \times d}$. Since the forward map is linear in $\mathbf{H}(\ell)$ at each layer, repeated application of the chain rule gives

$$\mathbf{J}_{is}(\ell) = \Big(\mathbf{M}(\ell - 1)\mathbf{M}(\ell - 2)\cdots\mathbf{M}(0)\Big)_{is} \cdot \Big(\mathbf{W}(0)\mathbf{W}(1)\cdots\mathbf{W}(\ell - 1)\Big), \tag{14}$$

with $\mathbf{J}_{is}(0) = \mathbf{I}_d$ when $i = s$ and $\mathbf{0}$ otherwise.

We now lower bound the scalar coefficient in (14). Because all entries of $\mathbf{M}(p)$ are nonnegative, for any $i, s \in \mathcal{C}_r$,

$$\Big(\mathbf{M}(\ell - 1)\cdots\mathbf{M}(0)\Big)_{is} \geq \Big(\mathbf{P}_\theta(\ell - 1)\cdots\mathbf{P}_\theta(0)\Big)_{is}.$$

Moreover, using (13), for every $p$ and every $u, v \in \mathcal{C}_r$, $\big(\mathbf{P}_\theta(p)\big)_{uv} = \alpha_{\theta,r}^{(p)}/n_r$, hence

$$\Big(\mathbf{P}_\theta(\ell - 1)\cdots\mathbf{P}_\theta(0)\Big)_{is} = \prod_{p=0}^{\ell-1} \frac{\alpha_{\theta,r}^{(p)}}{n_r}.$$

Combining the last two displays yields

$$\Big(\mathbf{M}(\ell - 1)\cdots\mathbf{M}(0)\Big)_{is} \geq \prod_{p=0}^{\ell-1} \frac{\alpha_{\theta,r}^{(p)}}{n_r}. \tag{15}$$

Finally, let

$$\mathbf{W}_{\text{tot}}(\ell) := \mathbf{W}(0)\mathbf{W}(1)\cdots\mathbf{W}(\ell - 1).$$

By the assumption $\sigma_{\min}(\mathbf{W}(p)) > 0$ for all $p$, we have $\sigma_{\min}(\mathbf{W}_{\text{tot}}(\ell)) \geq \prod_{p=0}^{\ell-1} \sigma_{\min}(\mathbf{W}(p))$.

Then, by (14) and (15),

$$\left\|\frac{\partial \mathbf{h}_i(\ell)}{\partial \mathbf{x}_s}\right\| = \|\mathbf{J}_{is}(\ell)\|$$

$$= \Big(\mathbf{M}(\ell - 1)\cdots\mathbf{M}(0)\Big)_{is} \cdot \|\mathbf{W}_{\text{tot}}(\ell)\| \geq \left(\prod_{p=0}^{\ell-1} \frac{\alpha_{\theta,r}^{(p)}}{n_r}\right)\left(\prod_{p=0}^{\ell-1} \sigma_{\min}\big(\mathbf{W}(p)\big)\right).$$

Multiplying the right-hand side by $\varepsilon^\ell$ (consistent with the $\varepsilon$-scaling in Eq. (9)) gives exactly (10). The lower bound depends only on $n_r$ and the per-layer parameters $\alpha_{\theta,r}^{(p)}$ and $\mathbf{W}(p)$, and is independent of the graph distance between $i$ and $s$ within $\mathcal{C}_r$. Furthermore, the lower bound for the model using different $\mathbf{W}$ for spatial and spectral parts can be obtained by following the same strategy, with additional consideration on the interactive terms between weight matrices; we omit it here. $\square$

*Proof of Proposition 2.* The proof follows the similar strategy in (Hariri et al., 2025). The core is the outer product between one antisymmetric matrix and a symmetric matrix is still antisymmetric, i.e, $\mathbf{P}_\theta \otimes \widehat{\mathbf{W}}$ and $\mathbf{A} \otimes \widehat{\mathbf{W}}$.

Let $\mathbf{h}(\ell) = \mathrm{vec}(\mathbf{H}(\ell)) \in \mathbb{R}^{Nd}$. Using the standard identity

$$\mathrm{vec}(\mathbf{S\,H\,W}) = (\mathbf{W}^\top \otimes \mathbf{S})\,\mathrm{vec}(\mathbf{H}),$$

the dynamics in Eq. (11) implies

$$
\begin{aligned}
\mathbf{h}(\ell+1) &= \mathbf{h}(\ell) + \epsilon\Big(\mathrm{vec}\big(\mathbf{P}_\theta(\ell)\mathbf{H}(\ell)\widehat{\mathbf{W}}(\ell)\big) + \mathrm{vec}\big(\widehat{\mathbf{A}}\mathbf{H}(\ell)\widehat{\mathbf{W}}(\ell)\big)\Big) \\
&= \mathbf{h}(\ell) + \epsilon\Big(\widehat{\mathbf{W}}(\ell)^\top \otimes \mathbf{P}_\theta(\ell) + \widehat{\mathbf{W}}(\ell)^\top \otimes \widehat{\mathbf{A}}\Big)\mathbf{h}(\ell) \\
&= \Big(\mathbf{I}_{Nd} + \epsilon\,\mathbf{A}_{\mathrm{tot}}(\ell)\Big)\mathbf{h}(\ell),
\end{aligned}
\tag{16}
$$

where

$$\mathbf{A}_{\mathrm{tot}}(\ell) := \widehat{\mathbf{W}}(\ell)^\top \otimes \big(\mathbf{P}_\theta(\ell) + \widehat{\mathbf{A}}\big) \in \mathbb{R}^{Nd \times Nd}.$$

Therefore the layer Jacobian with respect to $\mathbf{h}(\ell)$ is

$$\mathbf{J}(\ell) = \frac{\partial \mathbf{h}(\ell+1)}{\partial \mathbf{h}(\ell)} = \mathbf{I}_{Nd} + \epsilon\,\mathbf{A}_{\mathrm{tot}}(\ell).$$

We now show that $\mathbf{A}_{\mathrm{tot}}(\ell)$ is antisymmetric. By construction, $\widehat{\mathbf{W}}(\ell)$ is antisymmetric, i.e., $\widehat{\mathbf{W}}(\ell)^\top = -\widehat{\mathbf{W}}(\ell)$. Moreover, both $\mathbf{P}_\theta(\ell)$ and $\widehat{\mathbf{A}}$ are symmetric. Hence $\mathbf{P}_\theta(\ell) + \widehat{\mathbf{A}}$ is symmetric, and using $(\mathbf{X} \otimes \mathbf{Y})^\top = \mathbf{X}^\top \otimes \mathbf{Y}^\top$ gives

$$\mathbf{A}_{\mathrm{tot}}(\ell)^\top = \widehat{\mathbf{W}}(\ell) \otimes \big(\mathbf{P}_\theta(\ell) + \widehat{\mathbf{A}}\big) = -\widehat{\mathbf{W}}(\ell)^\top \otimes \big(\mathbf{P}_\theta(\ell) + \widehat{\mathbf{A}}\big) = -\mathbf{A}_{\mathrm{tot}}(\ell).$$

Thus $\mathbf{A}_{\mathrm{tot}}(\ell)$ is antisymmetric.

Finally,

$$
\begin{aligned}
\mathbf{J}(\ell)^\top \mathbf{J}(\ell) &= \big(\mathbf{I}_{Nd} + \epsilon\,\mathbf{A}_{\mathrm{tot}}(\ell)^\top\big)\big(\mathbf{I}_{Nd} + \epsilon\,\mathbf{A}_{\mathrm{tot}}(\ell)\big) \\
&= \mathbf{I}_{Nd} + \epsilon\big(\mathbf{A}_{\mathrm{tot}}(\ell) + \mathbf{A}_{\mathrm{tot}}(\ell)^\top\big) + \epsilon^2\,\mathbf{A}_{\mathrm{tot}}(\ell)^\top \mathbf{A}_{\mathrm{tot}}(\ell) \\
&= \mathbf{I}_{Nd} + \epsilon^2\,\mathbf{A}_{\mathrm{tot}}(\ell)^\top \mathbf{A}_{\mathrm{tot}}(\ell),
\end{aligned}
\tag{17}
$$

where the linear term vanishes since $\mathbf{A}_{\mathrm{tot}}(\ell)^\top = -\mathbf{A}_{\mathrm{tot}}(\ell)$. Because $\mathbf{A}_{\mathrm{tot}}(\ell)^\top \mathbf{A}_{\mathrm{tot}}(\ell)$ is symmetric positive semidefinite,

$$\|\mathbf{J}(\ell)\|_2^2 = \lambda_{\max}\big(\mathbf{J}(\ell)^\top \mathbf{J}(\ell)\big) = 1 + \epsilon^2\,\lambda_{\max}\big(\mathbf{A}_{\mathrm{tot}}(\ell)^\top \mathbf{A}_{\mathrm{tot}}(\ell)\big).$$

Taking square roots and using $\sqrt{1+x} = 1 + \mathcal{O}(x)$ as $x \to 0$ yields

$$\|\mathbf{J}(\ell)\|_2 = 1 + \mathcal{O}(\epsilon^2),$$

which proves the claimed layerwise Jacobian-energy stability. $\qquad\square$

## B. Experiment Details and Additional Analysis

### B.1. More Evidence on Jacobian Norm Distributions

To enrich the evidence on the Jacobian norm distribution difference between models, i.e., Figure 1, which only shows one sample randomly from the test set. In Figure 8, we additionally plot ten more results with the graph randomly picked up from the test set. These samples are with the node number ranging from 64 to 456. We found that, in all plots, S$^2$GNN's norm distributions are mixed **below-bound values** and **above-bound values**, whereas S$^3$GNNs are all above the theoretical lower bounds, except in the graph #356, where we observed around 5.7% of the norms are below the lower bound. These evidence further suggests the violation between the real-implementation and theoretical conclusions, and verifies the effectiveness of our proposed S$^3$GNN model.

### B.2. Baseline Description

We evaluate our proposed model by comparing it to various baselines. For standard graph learning tasks, including graph property prediction, large-scale node classification, and long-range graph learning, our baselines cover a broad spectrum of representative model families.

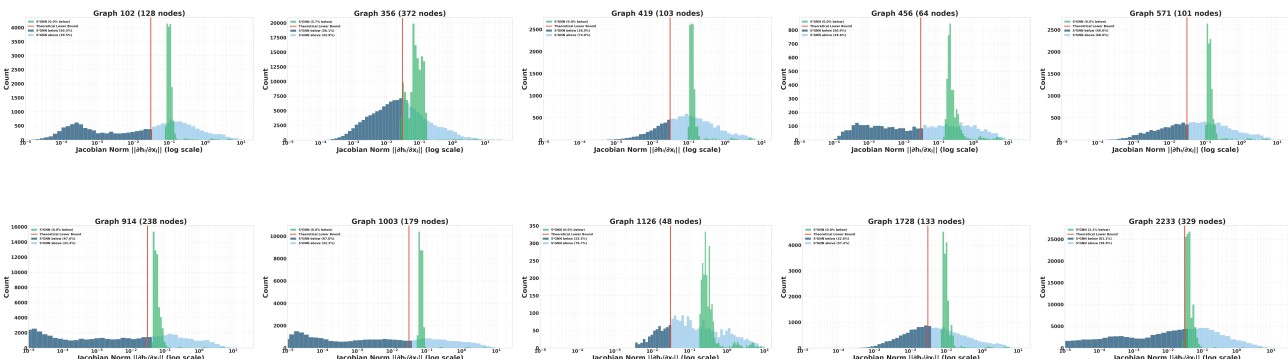

*Figure 8.* Additional evidence to show the differences in terms of Jacobian norm distributions between three models.

*Table 8.* Hyperparameters for LRGB Experiments

| Hyperparameter | Peptides-struct | Peptides-func | Search Space |
|---|---|---|---|
| Hidden dimension | 256 | 128 | 128, 256, 512 |
| Number of Layers | 5 | 7 | 3, 5, 7 |
| MLP Layers | 2 | 2 | 1, 2, 3 |
| Batch Size | 64 | 32 | 32, 64, 128 |
| Learning Rate | 0.001 | 0.001 | 0.0005, 0.001, 0.005 |
| Step Size ($\varepsilon$) | 0.5 | 0.1 | 0.1, 0.5, 1.0 |
| Initial $\alpha$ | 1.0 | 1.0 | 1, 2, 3 |
| Dissipation ($g$) | 0.1 | 0.1 | 0.01, 0.05, 0.1 |
| Dropout | 0.15 | 0.2 | 0.1, 0.15, 0.2 |
| Activation | SiLU | SiLU | SiLU, ReLU, tanh |
| Epochs | 200 | 200 | – |

**Graph Property Prediction.**    For graph-level property prediction tasks, we compare against classical and modern GNN baselines, including GCN (Kipf & Welling, 2017), GAT (Veličković et al., 2018), GraphSAGE (Hamilton et al., 2017), GIN (Xu et al., 2019), GRAND (Chamberlain et al., 2021b), A-DGN (Gravina et al., 2023), ChebNet (Defferrard et al., 2016), Stable-ChebNet (Hariri et al., 2025), DIFFormer (Wu et al., 2023a) and $S^2$GNN (Geisler et al., 2024).

**Node Classification.**    For large-scale node classification on OGB datasets, we compare against widely used and recent models, including GCN (Kipf & Welling, 2017), ChebNet (Defferrard et al., 2016), ChebNetII (He et al., 2022), Graph-SAGE (Hamilton et al., 2017), GAT (Veličković et al., 2018), NodeFormer (Wu et al., 2022), GraphGPS (Rampášek et al., 2022), GOAT (Kong et al., 2023), EXPHORMER+GCN (Shirzad et al., 2023), SPEXPHORMER (Shirzad et al., 2024), DIFFormer (Wu et al., 2023a), and Stable-ChebNet (Hariri et al., 2025).

**Long-Range Graph Learning.**    For long-range graph benchmarks, we include representative baselines from multiple model families. Specifically, we compare with transformer-based models including SAN+LapPE (Kreuzer et al., 2021), TIGT (Choi et al., 2024b), Specformer (Bo et al., 2023), Exphormer (Shirzad et al., 2024), G.MLPMixer (He et al., 2023), and GRIT (Ma et al., 2023); graph rewiring methods including LASER (Barbero et al., 2024) and DRew-GCN (Gutteridge et al., 2023); state-space models including Graph Mamba (Wang et al., 2024) and GMN (Behrouz & Hashemi, 2024).

### B.3. Hyperparameter Searching Space

In this section, we provide hyperparameter searching spaces for each of the experiments. Table 8 to 10 summarize the search spaces and the reference hyperparameters. We note that for the experiment in KGQA, topological interpolation, and the fulid dynamic prediction, the hyperparameters for S$^3$GNN are fixed with step size $\epsilon = 0.1$, Dissipative force $\gamma = 0.01$ and initial $\alpha = 1$, while we keep the rest of the settings exactly the same as the original paper. Please see (Mavromatis & Karypis, 2024; Yang, 2025; Pfaff et al., 2020) for more implementation details.

*Table 9.* Hyper-parameter grid for S³GNN on Graph Property Prediction tasks.

| Hyper-parameter | Diameter | SSSP | Eccentricity | Search Space |
|---|---|---|---|---|
| Hidden dim $d$ | 40 | 40 | 32 | 32, 40, 96 |
| Num of layers | 6 | 6 | 12 | 3, 6, 12, 24 |
| Step size $\varepsilon$ | 0.7 | 0.1 | 0.7 | 0.1, 1.0 |
| Dissipative force $\gamma$ | 0.01 | 0.01 | 0.01 | 0.001, 0.01, 0.05, 0.1 |
| $\alpha$ init | 1.0 | 1.0 | 1.0 | 0.5, 1.0, 2.0 |
| Batch size | 16 | 16 | 16 | 16, 32, 64 |
| Learning rate | 0.001 | 0.003 | 0.001 | 0.001, 0.003, 0.01 |
| Weight decay | $10^{-6}$ | $10^{-6}$ | $10^{-6}$ | $10^{-6}, 10^{-5}, 10^{-4}$ |
| Epochs | 1000 | 1000 | 1000 | 1000 |

*Table 10.* Hyperparameters for OGB Node Classification Experiments

| Hyper-parameter | OGBN-Proteins | OGBN-ArXiv | Search Space |
|---|---|---|---|
| Hidden Channels | 256 | 256 | 128, 256, 512, 1024 |
| Number of Layers | 10 | 5 | 2, 3, 4, 5, 7, 10 |
| Step Size ($\varepsilon$) | 0.1 | 0.1 | 0.1, 1.0 |
| Dissipation ($g$) | 0.1 | 0.1 | 0.01, 0.05, 0.1 |
| Initial $\alpha$ | 1.0 | 1.0 | 1, 2, 3 |
| Learning Rate | 0.001 | 0.001 | 0.0005, 0.001, 0.005, 0.01 |
| Batch Size | 1024 | 1024 | 256, 512, 1024 |
| Dropout | 0.1 | 0.2 | 0.1, 0.2, 0.3 |
| Activation | tanh | ReLU | tanh, ReLU |

## C. Performance on Heterophily Graphs

In this section, we evaluate S³GNN performance on heterophily graphs. We follow the settings in (Behrouz & Hashemi, 2024; Hariri et al., 2025). Specifically, four node classification tasks in the datasets of **Roman-empire**, **Amazon-ratings** and **Minesweeper**, and **Tolokers** are considered. We followed the standard data split for these datasets and the results are presented in Table 11, we note that some of the results are copied based on the online available results (Behrouz & Hashemi, 2024; Hariri et al., 2025). One can see that although with a stronger global mixing effect than classic GCN, causing better information transaction, S³GNN still shows state-of-the-art results for all datasets.

*Table 11.* Mean test set score and std averaged over 4 random weight initializations on heterophilic datasets. The higher, the better.

| Model | Roman-empire Acc↑ | Amazon-ratings Acc↑ | Minesweeper AUC↑ | Tolokers AUC↑ |
|---|---|---|---|---|
| MLP-2 | $66.04 \pm 0.71$ | $49.55 \pm 0.81$ | $50.92 \pm 1.25$ | $74.58 \pm 0.75$ |
| SGC-1 | $44.60 \pm 0.52$ | $40.69 \pm 0.42$ | $82.04 \pm 0.77$ | $73.80 \pm 1.35$ |
| MLP-1 | $64.12 \pm 0.61$ | $38.60 \pm 0.41$ | $50.59 \pm 0.83$ | $71.89 \pm 0.82$ |
| **MPNNs** | | | | |
| GAT | $80.87 \pm 0.30$ | $49.09 \pm 0.63$ | $92.01 \pm 0.68$ | $83.70 \pm 0.47$ |
| GAT (LapPE) | $84.80 \pm 0.46$ | $44.90 \pm 0.73$ | $93.50 \pm 0.54$ | $84.99 \pm 0.54$ |
| GAT (RWSE) | $86.62 \pm 0.53$ | $48.58 \pm 0.41$ | $92.53 \pm 0.65$ | $85.02 \pm 0.67$ |
| Gated-GCN | $74.46 \pm 0.54$ | $43.00 \pm 0.32$ | $87.54 \pm 1.22$ | $77.31 \pm 1.14$ |
| GCN | $73.69 \pm 0.74$ | $48.70 \pm 0.63$ | $89.75 \pm 0.52$ | $83.64 \pm 0.67$ |
| GCN (LapPE) | $83.37 \pm 0.55$ | $44.35 \pm 0.36$ | $94.26 \pm 0.49$ | $84.95 \pm 0.78$ |
| GCN (RWSE) | $84.84 \pm 0.55$ | $46.40 \pm 0.55$ | $93.84 \pm 0.48$ | $85.11 \pm 0.77$ |
| CO-GNN$(\Sigma, \Sigma)$ | $91.57 \pm 0.32$ | $51.28 \pm 0.56$ | $95.09 \pm 1.18$ | $83.36 \pm 0.89$ |
| CO-GNN$(\mu, \mu)$ | $91.37 \pm 0.35$ | $54.17 \pm 0.37$ | $97.31 \pm 0.41$ | $84.45 \pm 1.17$ |
| SAGE | $85.74 \pm 0.67$ | $53.63 \pm 0.39$ | $93.51 \pm 0.57$ | $82.43 \pm 0.44$ |
| **Graph Transformers** | | | | |
| Exphormer | $89.03 \pm 0.37$ | $53.51 \pm 0.46$ | $90.74 \pm 0.53$ | $83.77 \pm 0.78$ |
| NAGphormer | $74.34 \pm 0.77$ | $51.26 \pm 0.72$ | $84.19 \pm 0.66$ | $78.32 \pm 0.95$ |
| GOAT | $71.59 \pm 1.25$ | $44.61 \pm 0.50$ | $81.09 \pm 1.02$ | $83.11 \pm 1.04$ |
| GPS | $82.00 \pm 0.61$ | $53.10 \pm 0.42$ | $90.63 \pm 0.67$ | $83.71 \pm 0.48$ |
| GPS$_{\text{GCN+Performer}}$ (LapPE) | $83.96 \pm 0.53$ | $48.20 \pm 0.67$ | $93.85 \pm 0.41$ | $84.72 \pm 0.77$ |
| GPS$_{\text{GCN+Performer}}$ (RWSE) | $84.72 \pm 0.65$ | $48.08 \pm 0.85$ | $92.88 \pm 0.50$ | $84.81 \pm 0.86$ |
| GT | $86.51 \pm 0.73$ | $51.17 \pm 0.66$ | $91.85 \pm 0.76$ | $83.23 \pm 0.64$ |
| GT-sep | $87.32 \pm 0.39$ | $52.18 \pm 0.80$ | $92.29 \pm 0.47$ | $82.52 \pm 0.92$ |
| Polynomial | $\mathbf{92.55 \pm 0.30}$ | $\mathbf{54.81 \pm 0.49}$ | $97.46 \pm 0.36$ | $85.91 \pm 0.74$ |
| **Heterophily-Designated GNNs** | | | | |
| CPGNN | $63.96 \pm 0.62$ | $39.79 \pm 0.77$ | $52.03 \pm 5.46$ | $73.36 \pm 1.01$ |
| FAGCN | $65.22 \pm 0.56$ | $44.12 \pm 0.30$ | $88.17 \pm 0.73$ | $77.75 \pm 1.05$ |
| FSGNN | $79.92 \pm 0.56$ | $52.74 \pm 0.83$ | $90.08 \pm 0.70$ | $82.76 \pm 0.61$ |
| GBK-GNN | $74.57 \pm 0.47$ | $45.98 \pm 0.71$ | $90.85 \pm 0.58$ | $81.01 \pm 0.67$ |
| GloGNN | $59.63 \pm 0.69$ | $36.89 \pm 0.14$ | $51.08 \pm 1.23$ | $73.39 \pm 1.17$ |
| GPR-GNN | $64.85 \pm 0.27$ | $44.88 \pm 0.34$ | $86.24 \pm 0.61$ | $72.94 \pm 0.97$ |
| H2GCN | $60.11 \pm 0.52$ | $36.47 \pm 0.23$ | $89.71 \pm 0.31$ | $73.35 \pm 1.01$ |
| JacobiConv | $71.14 \pm 0.42$ | $43.55 \pm 0.48$ | $89.66 \pm 0.40$ | $68.66 \pm 0.65$ |
| **Graph SSMs** | | | | |
| GMN | $87.69 \pm 0.50$ | $54.07 \pm 0.31$ | $91.01 \pm 0.23$ | $84.52 \pm 0.21$ |
| GPS + Mamba | $83.10 \pm 0.28$ | $45.13 \pm 0.97$ | $89.93 \pm 0.54$ | $83.70 \pm 1.05$ |
| GRAMA$_{\text{GCN}}$ | $88.61 \pm 0.43$ | $53.48 \pm 0.62$ | $95.27 \pm 0.71$ | $\mathbf{86.23 \pm 1.10}$ |
| MP-SSM | $90.91 \pm 0.48$ | $53.65 \pm 0.71$ | $95.33 \pm 0.72$ | $85.26 \pm 0.93$ |
| S$^3$GNN(**ours**) | $91.52 \pm 0.15$ | $53.46 \pm 0.70$ | $\mathbf{97.58 \pm 0.44}$ | $85.89 \pm 1.05$ |

