# OpenReview forum: "S$^3$GNN: Efficient Global Mixing and Local Message Passing for Long-Range Graph Learning"
_ICML.cc/2026/Conference — ICML 2026 spotlight_

### Official Review · Reviewer_UXBp · 2026-03-07

**Soundness:** 3
**Presentation:** 3
**Significance:** 3
**Originality:** 2
**Overall Recommendation:** 5
**Confidence:** 4

**Summary:**

The authors revisit the assumptions made by S$^2$GNN and argue that achieving the Jacobian sensitivity lower bound is difficult in practice. They then propose S$^3$GNN, a global-mixing approach that avoids eigendecompositions for long-range information propagation, and instead couples sparse local aggregation with a cheap low-rank global mixing term designed to improve long-range communication while remaining scalable. The authors provide an extensive empirical evaluation of their method, showing gains in multiple scenarios.

**Compliance With Llm Reviewing Policy:**

Affirmed.

**Final Justification:**

All my concerns have been addressed.

**Key Questions For Authors:**

1. Can the authors compare against a simple virtual-node baseline?
2. Can the authors provide direct runtime and memory comparisons to S$^2$GNN and related spectral baselines?
3. Have the authors considered adding more long-range tasks, such as the synthetic and associative recall benchmarks used in S$^2$GNN?

**Limitations:**

No limitations or impact statements, but this is very minor in the context of the paper.

**Strengths And Weaknesses:**

**Strengths**:
1. The paper has a clear practical motivation: preserve the useful global-mixing intuition from S$^2$GNN, but remove the costly eigendecomposition from the implementation. This makes the proposed architecture substantially simpler and more scalable in principle.
2. The theoretical analysis is reasonably well aligned with the proposed model. In particular, Proposition 1 and Proposition 2 are not especially deep, but they are clearly connected to the architecture and help justify its design. The resulting story is coherent: the model aims to preserve non-vanishing long-range influence while stabilizing the Jacobian through constrained feature transforms.
3. The empirical section is broad and especially strong. The method performs well on graph property prediction, OGB node classification, and the LRGB, and the application experiments on KGQA and fluid dynamics are also promising.
4. The paper also offers a useful practical message: modelling long-range relationships may not require a fully spectral or attention-based architecture, and a simple global mixing mechanism can already be effective.


**Weaknesses**:
1. The critique of S$^2$GNN is somewhat overstated. As far as I understand, the original S$^2$GNN lower-bound result (Theorem 2 in the S$^2$GNN paper) is an existential statement under a particular parameter regime. Therefore, showing that the default practical implementation does not realize that regime does not really refute the original theorem; it mainly highlights a gap between theory and practice. I think this point is fair, but the paper occasionally presents it more aggressively than warranted.
2. The novelty relative to prior work feels somewhat incremental. S$^3$GNN can be viewed as combining the global zero-eigenspace mixing intuition of S$^2$GNN with the antisymmetric stabilization strategy used in Stable-ChebNet. I think this synthesis is useful, but the paper should position itself more clearly as a practical combination and simplification of existing ideas.
3. I am also not fully convinced by the long-range evaluation. Although the peptide benchmarks from the LRGB are commonly used in this setting, it remains unclear whether they truly require long-range capabilities (Toenshoff et al., "Where Did the Gap Go? Reassessing the Long-Range Graph Benchmark", LoG 2023). Adding the synthetic and mechanistic tasks used in S$^2$GNN, such as LR-CLUSTER, distance regression, oversquashing benchmarks, and associative recall, would have made the empirical validation more convincing.
4. The practical claim would be stronger with more direct efficiency comparisons. The paper argues that S$^3$GNN is cheaper than S$^2$GNN and spectral baselines, but I would have appreciated explicit wall-clock time and memory comparisons on the same hardware and datasets, rather than mainly asymptotic arguments and selected epoch times.
5. A more important empirical omission is the lack of comparison to simpler global baselines. In particular, a comparison against a simple virtual-node baseline seems necessary to understand how much of the gain really comes from the specific S$^3$GNN design.
6. The paper is also somewhat under-polished in presentation and occasionally hard to follow. There are several noticeable typos, awkward formulations, and some claims are phrased more strongly than the evidence strictly supports. The paper would also benefit from a more explicit discussion of its limitations. In addition, a few organizational choices hurt readability. For example:
	1. The enumeration on lines 184-199 in column 2 breaks the flow and is not self-contained, since the dataset and metric are only introduced in the previous paragraph;
	2. The results discussed on lines 158-161 would fit better in the experimental section;
	3. The questions on lines 145-148 in column 1 / 142-145 in column 2 should use a colon instead of a period;
	4. Typos: L152-153 col 1: "polling"; L177 col 1: "scaler"; L215-216 col 1: "constrain" etc.
These issues are not fatal, but they do affect the paper's overall clarity.
7.  I recommend that the authors include their figures as PDF/SVG files, such that they scale properly when zoomed. Figure 3 particularly looks very bad because it was resized vertically.  It is also weird to include a table as a figure (Fig. 2). Please consider using minipages or moving the table.
8. The proposed method is closely related to methods that propose or analyze virtual nodes for MPNNs. However, the related work doesn't discuss these techniques (e.g. Qian et al., "Probabilistic Graph Rewiring via Virtual Nodes", NeurIPS 2024; Southern et al., "Understanding Virtual Nodes: Oversquashing and Node Heterogeneity", ICLR 2025 etc.). I would encourage the authors to expand their discussion in the related work.

Overall, I think this is a useful paper with a clear practical motivation, a coherent model design story, and a broad empirical evaluation. While the novelty is somewhat incremental and some of the claims and comparisons could be improved, I lean towards a weak accept, mainly because the evaluation goes beyond standard graph benchmarks and includes more practical settings such as KGQA and fluid dynamics, which makes the paper more compelling from an applied perspective.

I would be open to increasing my score after the rebuttal if the authors expand the related work to explicitly cover virtual node papers, add comparisons to simple baselines such as a virtual node baseline, and improve the writing and presentation.

---

> ### Author Rebuttal · Authors · 2026-03-30
>
> Dear Reviewer UXBp,
>
> We thank the reviewer for the constructive feedback and positive recognition of our work. Below, we provide responses and additional results, such as virtual node baselines and a list of additional related works. We hope these revisions could adequately address your concerns.
>
> &nbsp;
>
> ## **[W1] Overstated critique of S2GNN**
> We agree that the lower bound in $S^2GNN$ is an existential result under specific parameter regimes. Our goal is not to refute this result, but to examine its realizability and provide a more practical solution. We will revise the wording accordingly to present our contribution in a more appropriate and precise manner.
>
> &nbsp;
>
> ## **[W2] Discussion of Novelty and Positioning**
> ### (1) Positioning and practical contribution
> We understand your concern regarding novelty and appreciate your positive recognition that this synthesis is useful. We would like to clarify that our method achieves several advantages over its original components. In the revised version, we will show these advantages more clearly. For example, compared with $S^2GNN$, our method avoids eigendecomposition and manual eigenspace selection; compared with Stable-ChebNet, it avoids high-order polynomial approximation for spectral filtering.
>
> &nbsp;
>
> ### (2) Strengthened theoretical guarantee and revision
> In addition, after re-examining Proposition 1, we believe the lower bound holds for an arbitrary unit vector $\mathbf u$ rather than some $\mathbf u$, since $||\mathbf W \mathbf u|| \geq \sigma_{\min}(\mathbf W)$ for any unit vector $\mathbf u$. This strengthens our claim, as $S^3GNN$ admits a positive lower bound on the Jacobian norm without requiring the idealized regime assumed in $S^2GNN$. We will clarify this in the revised paper.
>
> &nbsp;
>
> ## **[W3 & Q3] Synthetic and mechanistic long-range tasks**
> We have added several new experiments under the suggested settings. Two examples are shown below.
>
> ### (1) Results on LR-CLUSTER
> We simulated the LR-CLUSTER dataset following the configuration used in $S^2GNN$, which yields 12,000 graphs with an average of 896 nodes per graph. Due to word limit, here we only compare with $S^2GNN$, which is relatively powerful compared to other baselines. The results below show that our model outperforms $S^2GNN$ in all layer settings (3, 6, 12).
> |Layers (ℓ)|S²GNN|S³GNN|
> |-|-|-|
> |3|42.1 ± 1.02|**55.3 ± 0.90**|
> |6|61.3 ± 0.47|**64.8 ± 0.86**|
> |12|81.6 ± 0.78|**84.2 ± 0.65**|
>
> &nbsp;
>
> ### (2) Results on Oversquashing-extend
> Following $S^2GNN$, we simulated the oversquashing-extend dataset, which contains 730 graphs (Clique path + Ring Graph) with an average of 43.8 nodes per graph. As shown below, our method outperforms $S^2GNN$.
> |Model/dataset|Clique Path+Ring Graph|
> |-|-|
> |S²GNN|95.1 ± 0.75|
> |**S³GNN**|**96.9 ± 0.33**|
>
> &nbsp;
>
> ## **[W4 & Q2] Runtime and Memory Comparison**
> We show below the comparison between our model and $S^2GNN$ in Peptides-Structural dataset. Both models are aligned to 500K parameters. We will extend this table to more baselines and tasks in the revised paper.
> |Model/dataset|Parameters|Total Time (200 epochs)|GPU (H200 SXM) Consumption|
> |-|-|-|-|
> |S²GNN|~500K|2944.8s|3.8GB|
> | **S³GNN**|**~500K**|**610.4s**|**1.5GB**|
>
> &nbsp;
>
> ## **[W5 & Q1] Comparison with Virtual-Node Baselines**
> Due to word limit, please refer to our response to **Reviewer SeNv [W1]** for experiment results and additional discussions of virtual-node baselines.
>
> &nbsp;
>
> ## **[W6 & W7] Presentation, Clarity, and Formatting**
>
> We will make the following improvements:
> - **W6**: Proofread to fix typos and improve clarity, revise organization, move misplaced results, and soften strong claims while clarifying limitations.
> - **W7**: Convert figures to vector formats (PDF/SVG), fix distortion in Fig. 3, and convert Fig. 2 into a proper table.
>
> &nbsp;
>
> ## **[W8] Related Work on Virtual Nodes**
> After a careful review of the literature, we will expand the related work section to include both classic and recently developed virtual node methods, including Gilmer et al. (2017), Qian et al. (2024), Rosenbluth et al. (2024), Southern et al. (2025), and Choi et al. (2026).
>
> In the revised paper, we will introduce a subsection under the Related Work, titled “Comparison with Virtual Node Methods,” to systematically discuss these works and clarify their relation to our method.
>
> &nbsp;
>
> ## **[Limitation]**
> In the revised manuscript, we will include a limitations section to explicitly discuss the scope and potential constraints of our method.
>
> &nbsp;
>
> ## **[References]**
> Choi et al. Are graph transformers necessary? Efficient long-range message passing with fractal nodes in MPNNs. 2026
>
> Gilmer et al. Neural message passing for quantum chemistry. 2017
>
> Qian et al. Probabilistic graph rewiring via virtual nodes. 2024
>
> Rosenbluth et al. Distinguished in uniform: Self attention vs. virtual nodes. 2024
>
> Southern et al. Understanding virtual nodes: Oversquashing and node heterogeneity. 2025

---

> > ### Author Rebuttal · Reviewer_UXBp · 2026-04-02
> >
> > I would like to thank the authors for the response, my concerns have been addressed, therefore I am raising my score to Accept.

---

> > > ### Author Response · Authors · 2026-04-02
> > >
> > > Dear Reviewer UXBp,
> > >
> > > Thank you very much for your acknowledgement. We sincerely appreciate the time and effort you devoted to reading our rebuttal and providing this confirmation. We are very grateful that our response has adequately addressed your concerns, and we truly appreciate your reconsideration of our work and the increase in score.
> > >
> > > Best regards,
> > > The Authors

---

### Official Review · Reviewer_SfLa · 2026-03-12

**Soundness:** 3
**Presentation:** 3
**Significance:** 2
**Originality:** 3
**Overall Recommendation:** 4
**Confidence:** 3

**Summary:**

$S^3GNN$ (Simple, Stable, and Spectral GNN) addresses the oversquashing (OSQ) problem in Graph Neural Networks, where information from distant nodes is compressed and lost during message passing. The paper identifies a significant gap between existing OSQ-mitigation theories and their practical implementations, noting that many theoretical lower bounds are difficult to attain without sacrificing efficiency or expressivity. The model reintroduces spatial message passing alongside a low-rank global mixing term. This design allows for efficient global information exchange without the $\mathcal{O}(N^3)$ cost of eigendecomposition or complex attention mechanisms.

**Compliance With Llm Reviewing Policy:**

Affirmed.

**Final Justification:**

My concerns are addressed. I will maintain my score.

**Key Questions For Authors:**

1. "The use of a low-rank global mixing term in $S^3GNN$  shares a high-level conceptual similarity with the low-rank global label relationship matrix used in the paper 'Predicting Global Label Relationship Matrix for Graph Neural Networks under Heterophily'.  Could you also discuss this related paper?

**Limitations:**

There is no Limitations section in the paper.

**Strengths And Weaknesses:**

### Strengths

1. Unlike many spectral GNNs that demand an $\mathcal{O}(N^3)$ eigendecomposition, $S^3GNN$ uses a low-rank global mixing approach. This allows it to achieve a per-layer complexity of $\mathcal{O}(|\mathcal{E}|d+Nd^{2})$, which is effectively as fast as a standard GCN. It provides the benefits of "global" information without the massive computational overhead usually associated with spectral methods.

2. It provides a mathematically proven, distance-independent lower bound for node influence. In practice, this translated to an order-of-magnitude error reduction on tasks like Eccentricity compared to previous state-of-the-art models like $S^2GNN$.

3. The model proved to be a "Swiss Army knife" in testing. It served as a high-performance backbone for everything from reasoning in Knowledge Graphs (KGQA) to predicting complex fluid dynamics in mesh-based simulations.

### Weaknesses
1. While $S^3GNN$ is efficient, its performance relies on carefully tuning parameters like the step size ($\epsilon$), the mixing coefficient ($\alpha$), and the "dissipative force" ($\gamma$). As seen in the appendix, different tasks (e.g., Peptides-struct vs. Peptides-func) required notably different configurations to reach peak accuracy

2. In some Retrieval-Augmented Generation (RAG) experiments for KGQA, the improvement over existing GNN-RAG pipelines was relatively small.

---

> ### Author Rebuttal · Authors · 2026-03-30
>
> Dear Reviewer SfLa,
>
> We sincerely thank you for your valuable feedback. In particular, we appreciate your observation regarding the performance in the RAG setting. This led us to conduct a deeper analysis of this phenomenon and provided useful guidance for strengthening the experimental discussion in the revised manuscript. In the following, we will clarify the hyperparameter tuning overhead, provide further analysis of the RAG results and discuss the relation to the existing low-rank approach. We hope these clarifications and revisions help address your concerns.
>
> &nbsp;
>
> ## **[W1] Hyperparameter Tuning**
> We appreciate the reviewer’s concern, but would like to respectfully clarify that this is not the case.
>
> In our experiments, $S^3GNN$ does not rely on careful task-specific tuning: we use the same hyperparameter search space as Stable-ChebNet, rather than introducing a more extensive or specially designed search for our method. In addition, the newly introduced mixing coefficient $\alpha$ does not appear to require delicate tuning in practice. As discussed in the “$\alpha$ Dynamics” (Section 4.1), the $\alpha$ values consistently increase with the number of layers and automatically satisfy the non-negative assumption during training, suggesting that this behavior is driven by the learned dynamics rather than sophisticated initialization.
>
> Therefore, while the best configurations may differ across tasks, this is standard in related works and does not indicate that $S^3GNN$ depends on unusually heavy tuning.
>
> &nbsp;
>
> ## **[W2] Limited Gains in RAG Setting**
> We thank the reviewer for this observation. We would like to clarify that the relatively smaller improvement in the RAG setting does not necessarily indicate a limitation of $S^3GNN$, but is more likely a consequence of the experimental setting itself.
>
> First, in our experiments, we only replace the reasoning GNN with $S^3GNN$ while keeping the rest of the RAG pipeline unchanged, so the observed gains reflect improvements in the reasoning module alone and may not transfer into equally large gains at the full-pipeline level.
>
> Second, the main advantage of $S^3GNN$ is less visible in RAG tasks as RAG has already alleviated part of the oversquashing problem by retrieving a _compact_ and relevant subgraph prior to reasoning. As shown in Table 4, $S^3GNN$ without RAG already matches or exceeds GNN-RAG in some settings (e.g., CWQ multi-hop), suggesting that $S^3GNN$’s dynamic is a potential substitute for RAG’s subgraph retrieval in handling the KGQA problem. As a result, the combination of these two techniques may not yield large additive gains, although consistent improvements can still be observed.
>
> In our revised paper, we will include the discussion of this phenomenon with the contents above.
>
> &nbsp;
>
> ## **[Q1] Relation to Prior Work**
> We thank the reviewer for mentioning this related work. We agree that both methods share a common high-level view: they use a low-rank global operator to enable information mixing among groups of nodes. In this sense, the two methods are conceptually related. However, the source of the low-rank structure is fundamentally different. In their work, the low rank matrix is an estimated global label-relationship matrix, so the grouping is label semantic and task dependent. In our method, the low-rank operator is induced directly by the graph’s 0-eigenspace, which corresponds to connected component level structural grouping. Therefore, our method may be viewed as a simpler and more structural form of low-rank global mixing, whereas their method is a label-aware low-rank relation learning approach.
>
> &nbsp;
>
> ## **[Limitation]**
> In the revised manuscript, we will include a limitations section to explicitly discuss the scope and potential constraints of our method.

---

> > ### Author Rebuttal · Reviewer_SfLa · 2026-04-01
> >
> > I don't have follow-up questions and will maintain my score.

---

> > > ### Author Response · Authors · 2026-04-02
> > >
> > > Dear Reviewer SfLa,
> > >
> > > Thank you very much for your acknowledgement and for the time and care you devoted to reading our rebuttal and considering our responses. We are very grateful that our clarifications have adequately addressed your concerns, and we sincerely appreciate your thoughtful evaluation of our work.
> > >
> > > Authors.

---

### Official Review · Reviewer_SeNv · 2026-03-14

**Soundness:** 3
**Presentation:** 3
**Significance:** 2
**Originality:** 2
**Overall Recommendation:** 4
**Confidence:** 3

**Summary:**

This paper proposes S3GNN, a graph neural network architecture designed to address the oversquashing problem in long-range graph learning. The model combines local message passing with a lightweight global mixing term derived from the projector onto the zero-eigenspace of the Laplacian. The authors argue that this design preserves long-range influence without requiring costly eigendecomposition, and they further introduce antisymmetric feature transformations to stabilize Jacobian dynamics across layers.

**Compliance With Llm Reviewing Policy:**

Affirmed.

**Final Justification:**

Given the responses of the authors during the rebuttal period, I updated my score from 3 to 4.

**Key Questions For Authors:**

See weaknesses.

**Limitations:**

No explicit discussion of limitations.

**Strengths And Weaknesses:**

strengths:

1. A strong aspect of the paper is the diagnostic analysis of the gap between the theoretical lower bound used in prior work and the behavior of the practical model implementation. The paper shows that the lower bound in Eq. (6) relies on conditions that differ substantially from the setting of the implemented model in Eq. (5). Figure 1 illustrates this clearly by plotting the distribution of Jacobian norms.

2. The proposed architecture is also relatively simple. In a connected graph, the global mixing component reduces to broadcasting a scaled average of node features across the graph. This avoids eigendecomposition, attention mechanisms, and other expensive spectral operations. The resulting computational complexity per layer is $O(|E|d + Nd^2)$, which is comparable to standard GCN-style architectures.

3. The breadth of the experimental evaluation. The paper evaluates the method across multiple settings, including long-range graph benchmarks, knowledge graph question answering, topological interpolation, and mesh-based fluid simulation.

weaknesses:
1. A primary concern is the novelty of the proposed global mixing mechanism.  For graphs with a single connected component, the global term effectively reduces to broadcasting a scaled global average of node features, i.e., a term proportional to $\mathbf{1}\mathbf{1}^\top H^{(\ell)}$.  This operation is closely related to the widely used \emph{virtual node} mechanism, where a global node aggregates and redistributes information across the graph.  However, the paper neither compares against virtual-node baselines nor discusses this connection in detail. As a result, it remains unclear whether the empirical gains come from the specific S3GNN formulation or simply from introducing global averaging.

2. The theoretical guarantee is relatively weak.  Proposition~1 only shows that there exists a direction in which the influence between nodes remains nonvanishing.  However, this does not imply that useful task-relevant information can propagate across layers.  In high-dimensional representations, preserving influence in a single direction may still allow most informative signals to vanish.

3. The sizeable improvements on synthetic tasts (like Barbell) do not transfer to real-world benchmarks (peptides-func, peptides-struct), where the improvements are marginal.

---

> ### Author Rebuttal · Authors · 2026-03-30
>
> Dear Reviewer SeNv,
>
> We sincerely thank you for constructive and insightful comments, particularly on our theoretical guarantee, which helped us identify an area for improvement and led us to strengthen our theoretical foundation. Below, we will clarify distinctions from prior work, refine our theoretical analysis, and expand the empirical evaluation with additional virtual node baselines. We hope our responses and revisions could adequately address your concerns.
>
> &nbsp;
>
> ## **[W1] Novelty of Global Mixing & Relation to Virtual Nodes**
> We agree that our method shares a similar high-level motivation with virtual-node-style methods, in the sense that both introduce an explicit global communication mechanism for long-range information exchange.
>
> &nbsp;
>
> ### (1) Clarification of mechanism difference
> $S^3GNN$ differs from standard virtual node (VN) methods and fractal node methods primarily in how the global interaction mechanism is defined. Existing VN-based methods typically rely on initial graph summarisation. For example, the VN in Gilmer et al. (2017) and Rosenbluth et al. (2024) receives graph-level information from an initial message-passing network with a readout layer; the fractal node method developed in Choi et al. (2026) relies on assessing the graph's original topological information. Instead, $S^3GNN$ only requires simple scaling of graph-connected component. In the revised paper, we will expand discussion of these differences in the related work section, which will be moved to the main text.
>
> &nbsp;
>
> ### (2) Empirical comparison with virtual node baselines
> We will incorporate several VN–based baselines into our experiments, including GCN+VN and GatedGCN+VN (Rosenbluth et al., 2024; Southern et al., 2025), as well as GCN+FN and GatedGCN+FN (Choi et al., 2026). For illustration, we present their results on the LRGB benchmark below.
>
> The results suggest that our model consistently outperforms classical VN-based methods. In the revised manuscript, we will further expand the comparison to additional settings for a more comprehensive evaluation.
> |Model / Dataset|Peptides-func (AP)|Peptides-struct (MAE)|
> |-|-|-|
> |GCN+VN (Rosenbluth et al., 2024)|65.4 ± 0.66|0.2509 ± 0.0030|
> |GatedGCN+VN (Rosenbluth et al., 2024)|68.8 ± 0.47|0.2460 ± 0.0021|
> |GatedGCN+VN (Southern et al., 2025)|68.1 ± 0.33|0.2498 ± 0.0042|
> |GCN+FN (Choi et al., 2026)|67.8 ± 0.54|0.2456 ± 0.0019|
> |GatedGCN+FN (Choi et al., 2026)|69.5 ± 0.47|0.2470 ± 0.0039|
> |**S³GNN (ours)**|**73.2 ± 0.14**|**0.2429 ± 0.0014**|
>
> &nbsp;
>
> ## **[W2] Theoretical Guarantee and Practical Effectiveness**
> ### (1) Clarification of proposition
> Upon re-examination, we believe the lower bound holds for an arbitrary unit vector $\mathbf u$ rather than for some $\mathbf u$, since $||\mathbf W \mathbf u|| \geq \sigma_{\min}(\mathbf W)$ for any unit vector $\mathbf u$. We will revise the proposition accordingly and update the rest of the paper, strengthening our theoretical contribution and improving alignment between our theory and empirical observations (e.g., Figure 1).
>
> &nbsp;
>
> ### (2) Empirical evidence
> Regarding the concern about the high-dimensional feature information propagation, we would like to clarify that we indeed have several experiments with high-dimensional node features, such as OGBN-Proteins (10 layers)  and OGBN-ArXiv (5 layers), both with hidden dimension 256; Peptides-struct (5 layers) and Peptides-func (7 layers), with the hidden dimension 256 and 128, respectively. These results show that the non-vanishing influence characterized by Proposition 1 is not limited to low-layer or low-dimensional scenarios, but remains meaningful in more complex tasks. We will add further discussion in the revised paper.
>
> &nbsp;
>
> ## **[W3] Synthetic Datasets to Real-World Benchmarks**
> We thank the reviewer for this observation. We would like to clarify that Barbell is a synthetic task primarily designed to test robustness to oversquashing, so the large gain reflects effectiveness on addressing this bottleneck. In contrast, real-world benchmarks such as Peptides-func and Peptides-struct are noisier and more complex, with additional sources of challenges beyond OSQ alone. As a result, improvements in long-range communication, which is our main contribution, may not translate into equally large gains on real data. Smaller but consistent improvements are therefore expected and still demonstrate practical value.
>
> &nbsp;
>
> ## **[Limitation]**
> In the revised manuscript, we will include a limitations section to explicitly discuss the scope and potential constraints of our method.
>
> &nbsp;
>
> ## **[References]**
> Choi et al. Are graph transformers necessary? Efficient long-range message passing with fractal nodes in MPNNs. 2026
>
> Gilmer et al. Neural message passing for quantum chemistry. 2017
>
> Rosenbluth et al. Distinguished in uniform: Self attention vs. virtual nodes. 2024
>
> Southern et al. Understanding virtual nodes: Oversquashing and node heterogeneity. 2025

---

> > ### Author Rebuttal · Reviewer_SeNv · 2026-04-01
> >
> > Thanks for the clear responses. I am a bit concerned with the phrasing:
> > "Upon re-examination, we believe the lower bound holds for an arbitrary unit vector ..."
> >
> > This is about a mathematical proof, there is nothing to "believe". It is either true or false.
> > I assume that this just has to do with the form of communicating rather than this being a conjecture. Is that correct?

---

> > > ### Author Response · Authors · 2026-04-01
> > >
> > > We agree that the wording was poor, but our conclusion is unchanged: the lower bound holds for any unit vector. We will revise the text accordingly. If this addresses your concern, we would appreciate your reconsideration of the score.

---

### Decision · Program_Chairs · 2026-04-30

**Decision:**

Accept (spotlight)

**Comment:**

The reviewers were overall enthusiastic about the paper, including its clear motivation, model design and evaluation. Specifically, reviewers highlighted the analysis of the gap between theoretical lower bounds of past approaches and existing implementations. They noted that the design of the model is relatively simple and coherent, and it is more scalable compared to spectral or attention-based approaches, with respect to runtime and memory usage. They also appreciated the breadth of experiments conducted across various datasets and task types, which demonstrates the model's versatility and robustness. Experiments comparing to VN-based models as well as additional long-range tasks (from S^2GNN) resolved some concerns.

However, there are some weaknesses and clarifications that need to be addressed in the final version. As the reviewers noted, some experiments demonstrated marginal performance which the authors explained in the rebuttal - this needs to be added to the paper. Missing limitations need to be added for a comprehensive view of the model's shortcomings. Also, clarity concerns and figure quality issues raised by reviewer UXBp should be addressed to improve the readability of the work. Finally,as requested by reviewer UXBp the authors need to explicitly position their contributions with respect to S^2GNN and Stable Cheb-Net.